# Statistical mechanics of coupled supercooled liquids in finite dimensions

Benjamin Guiselin,[1, *] Ludovic Berthier,[1, 2] and Gilles Tarjus[3]

[1]*Laboratoire Charles Coulomb (L2C), Université de Montpellier, CNRS, 34095 Montpellier, France*
[2]*Department of Chemistry, University of Cambridge,*
*Lensfield Road, Cambridge CB2 1EW, United Kingdom*
[3]*LPTMC, CNRS-UMR 7600, Sorbonne Université, 4 Pl. Jussieu, F-75005 Paris, France*
(Dated: January 28, 2022)

We study the statistical mechanics of supercooled liquids when the system evolves at a temperature $T$ with a field $\epsilon$ linearly coupled to its overlap with a reference configuration of the same liquid sampled at a temperature $T_0$. We use mean-field theory to fully characterize the influence of the reference temperature $T_0$, and we mainly study the case of a fixed, low-$T_0$ value in computer simulations. We numerically investigate the extended phase diagram in the $(\epsilon, T)$ plane of model glass-forming liquids in spatial dimensions $d = 2$ and $d = 3$, relying on umbrella sampling and reweighting techniques. For both $2d$ and $3d$ cases, a similar phenomenology with nontrivial thermodynamic fluctuations of the overlap is observed at low temperatures, but a detailed finite-size analysis reveals qualitatively distinct behaviors. We establish the existence of a first-order transition line for nonzero $\epsilon$ ending in a critical point in the universality class of the random-field Ising model (RFIM) in $d = 3$. In $d = 2$ instead, no phase transition is found in large enough systems at least down to temperatures below the extrapolated calorimetric glass transition temperature $T_g$. Our results confirm that glass-forming liquid samples of limited size display the thermodynamic fluctuations expected for finite systems undergoing a random first-order transition. They also support the relevance of the physics of the RFIM for supercooled liquids, which may then explain the qualitative difference between $2d$ and $3d$ glass-formers.

## I. INTRODUCTION

Glass formation is the direct consequence of the rapid evolution of dynamic properties of supercooled liquids as the temperature is decreased toward the experimental glass transition temperature $T_g$ [1, 2]. It is thus conceivable to explain this phenomenon by using kinetic concepts to directly account for slow molecular motion [3], such as free volume [4], kinetic constraints [5], or local barriers controlled by elasticity [6]. Yet, slow dynamics can also be regarded as an emerging physical property slaved to some important changes in static properties of the supercooled liquid [7], captured for instance by the evolution of the potential [8] and free-energy landscapes [9, 10], or geometric frustration [11]. In the mean-field limit or in large spatial dimensions [12], the evolution of the free-energy landscape directly reflects the approach to a random first-order transition (RFOT) to an ideal glass phase [9] that is accompanied by a vanishing configurational entropy at a "Kauzmann transition" temperature $T_K$. The emergence of metastable minima (states) in the free-energy landscape that can trap the system for increasingly long times is responsible for ergodicity breaking [13, 14]. The present work belongs to a large research effort to understand how finite-dimensional fluctuations affect this mean-field theoretical construction.

An elegant way to follow the evolution of the free-energy landscape of glass-formers as temperature is lowered was proposed long ago by Franz and Parisi [15–17]

and has since given rise to many studies [18–25]. It relies on studying the equilibrium statistical mechanics of a glass-forming liquid at a temperature $T$ in the presence of a finite attraction of amplitude $\epsilon$ to a quenched reference configuration of the same liquid sampled from the equilibrium Boltzmann distribution at a temperature $T_0$. In other words, one now studies the thermodynamics of a liquid in the presence of an imposed quenched disorder represented by the reference configuration, which will be called below a "constrained liquid": see the sketch in Fig. 1(a). (The annealed version, where both copies evolve simultaneously with an attraction, has also been studied [26–32].) In this construction, the similarity or overlap between the two copies (or replicas) is computed from the positions of the $N$ particles in the constrained liquid, denoted by $\boldsymbol{r}^N$, and the reference configuration, $\boldsymbol{r}_0^N$, as follows:

$$\widehat{Q}[\boldsymbol{r}^N; \boldsymbol{r}_0^N] = \frac{1}{N} \sum_{i,j=1}^{N} w(|\boldsymbol{r}_i - \boldsymbol{r}_{0,j}|/a). \tag{1}$$

The overlap represents the order parameter that distinguishes between a delocalized (liquid) phase of typical overlap $Q_{\text{rand}} \ll 1$ and a localized (glassy) phase with a large overlap $Q_g \approx 1$. In the above equation, $w(x)$ is a window function decreasing from 1 to 0 on a scale of order 1 and $a$ is a tolerance length that accounts for thermal vibrations in the localized phase [33].

An attractive coupling between the two copies is implemented by linearly biasing the overlap between the two configurations by means of a "source" $\epsilon$ in order to

* benjamin.guiselin@umontpellier.fr

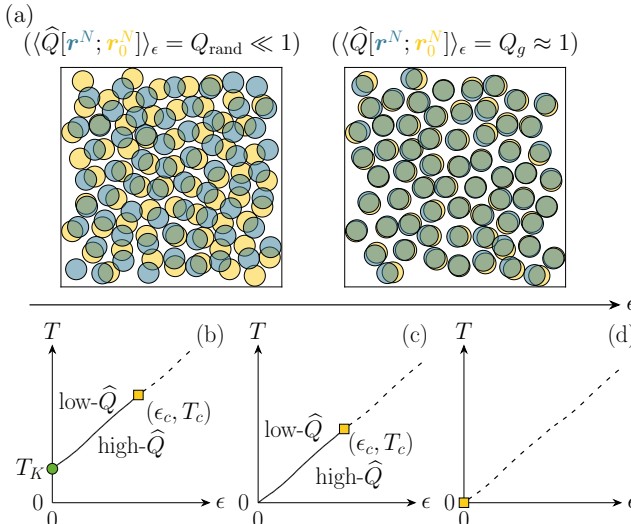

(a)
$(\langle\widehat{Q}[\boldsymbol{r}^N;\boldsymbol{r}_0^N]\rangle_\epsilon = Q_{\rm rand} \ll 1)$   $(\langle\widehat{Q}[\boldsymbol{r}^N;\boldsymbol{r}_0^N]\rangle_\epsilon = Q_g \approx 1)$

FIG. 1. (a) Glass-forming liquid (blue) evolving at a temperature $T$: its overlap with a quenched reference configuration (yellow) sampled at a temperature $T_0$ is linearly coupled to a source $\epsilon > 0$, which may trigger a transition from (left) a low-overlap delocalized (liquid) phase to (right) a large-overlap localized (glass) phase. (b, c) Possible phase diagrams for a constrained liquid with $T_0 = T$, depending on the existence [panel (b)] or the absence [panel (c)] of an entropy crisis and a random first-order transition at a nonzero $T_K$. Both diagrams display a line of first-order transition (full line) between low- and high-overlap phases ending in a critical point $(\epsilon_c, T_c)$. Above the critical point, a "Widom line" (dashed line) where the fluctuations of the overlap are maximum exists. (d) Possible phase diagram with no transition and a Widom line.

favor large overlap values when $\epsilon > 0$. If we let

$$\widehat{H}[\boldsymbol{r}^N] = \frac{1}{2}\sum_{i<j} v(|\boldsymbol{r}_i - \boldsymbol{r}_j|) \qquad (2)$$

denote the Hamiltonian of the unconstrained liquid with a pair interaction $v(r)$, the Hamiltonian of the liquid coupled to the reference configuration $\boldsymbol{r}_0^N$ reads

$$\widehat{H}_\epsilon[\boldsymbol{r}^N;\boldsymbol{r}_0^N] = \widehat{H}[\boldsymbol{r}^N] - N\epsilon\widehat{Q}[\boldsymbol{r}^N;\boldsymbol{r}_0^N], \qquad (3)$$

which defines the statistical-mechanical problem to be studied. The positions $\boldsymbol{r}_0^N$ in the reference configuration act as a source of quenched disorder for the Hamiltonian $\widehat{H}_\epsilon[\boldsymbol{r}^N;\boldsymbol{r}_0^N]$.

At a fixed temperature $T$, the state of the constrained liquid is obtained by minimizing its free energy. Qualitatively, at low $\epsilon$, entropy dominates and can be maximized by a full exploration of the configuration space. The system is thus a delocalized liquid, which is never close to the reference configuration and the overlap is small. Instead, at large $\epsilon$, the attraction energy dominates and the system acquires a large overlap with the reference configuration by staying very close to it. The system is then in a localized glass phase. The Franz-Parisi construction with a source $\epsilon$ allows one to track the evolution between

these two regimes and how it may lead, in the thermodynamic limit, to an equilibrium phase transition [16].

Investigating equilibrium phase transitions for constrained liquids is valuable for several reasons. First, they give insight into the statistical properties of the underlying landscape characterizing glass-formers and indicate whether localized glassy states exist in the system. The existence of phase transitions then suggests that, in the unconstrained liquid, one can meaningfully define a glass phase, which is metastable with respect to the liquid phase (for $T > T_K$). On the other hand, the absence of such phase transitions implies, *inter alia*, the absence of a thermodynamic glass transition (RFOT). As such, this provides a complementary tool to other approaches such as measurements of the point-to-set length [34–37] and of the configurational entropy [38–41]. Second, because they may take place at temperatures and conditions under which the glassy slowdown of relaxation is not too severe, they can be directly observed in equilibrium conditions, rather than extrapolated as the ideal glass transition; they can furthermore be crisply defined, unlike, *e.g.*, the dynamical mode-coupling crossover [42].

The thermodynamics of constrained liquids can be computed exactly in the limit of infinite dimensions [43], which is equivalent to a mean-field treatment [12]. Possible phase diagrams in the $(\epsilon, T)$ plane are shown in Fig. 1(b)-(c) for the case where the reference configurations are sampled at the same temperature $T$ as the constrained liquid [44], *i.e.*, $T_0 = T$. They both display a first-order transition line separating the localized and delocalized phases, ending in a critical point at $(\epsilon_c, T_c)$. This line may either converge at low $T$ to $(0, T_K > 0)$ if the system has a vanishing configurational entropy (Kauzmann transition) at a nonzero $T_K$, or to $(0, 0)$ if not. On the first-order transition line and at the critical point, the variance of the properly defined overlap fluctuations diverges in the thermodynamic limit. In contrast, above the critical point, the variance of the overlap fluctuations stays finite for any $\epsilon$. It displays a maximum at fixed $T$, which defines the so-called "Widom line" [45]. While a nonzero $T_K$ implies the existence of a second-order critical point at $(\epsilon_c, T_c)$ [see Fig. 1(b)], the reverse is not true [see Fig. 1(c)].

Extrapolating the physics of glass formation from $d = \infty$ down to $d = 2, 3$ is nontrivial due to finite-dimensional fluctuations which can have a dramatic effect on mean-field constructs such as metastable states. There is no guarantee, then, that any of the mean-field predictions survives in the $(\epsilon, T)$ phase diagram: one may rather find something as sketched in Fig. 1(d), with no singularity and just a Widom line going down to zero temperature. (Of course, the phase diagrams displayed in Fig. 1 do not exhaust all possibilities: see, *e.g.*, Ref. [44].)

Recent field-theoretical calculations based on an effective description in terms of a Landau-Ginzburg free-energy functional of the overlap have shown that a constrained glass-forming liquid close to its putative critical point $(\epsilon_c, T_c)$ can be mapped onto a disordered system

described by a $\phi^4$-theory in the presence of a random field [21, 22]. This shows that if the critical point survives in finite $d$, it should be in the universality class of the random-field Ising model (RFIM) [46]. Modulo some adjustments, this mapping applies to the first-order transition line [47, 48]. This result also implies that there should be no transition in $2d$, whatever $\epsilon$, because $d = 2$ is the lower critical dimension of the RFIM [49, 50]. For $2d$ glass-forming liquids, we may thus postulate a phase diagram as in Fig. 1(d). In contrast, the transition in $3d$ may survive if the strength of the effective random field is not too large [46, 51, 52], and phase diagrams as illustrated in Fig. 1(b)-(c) could then be expected.

The Hamiltonian in Eq. (3) has been the subject of a number of numerical analyses for both models of atomic liquids [17, 19, 20, 23, 24, 26] and spin plaquette models [53]. Early studies suffered from sampling issues which were later solved by introducing biased sampling techniques and reweighting methods [26]. All studies on $3d$ constrained systems pointed to the existence of a phase transition and RFIM-like behavior, but accessing large system sizes for model glass-forming liquids was not possible. By combining the biased sampling and reweighting techniques with the accelerated exploration of the configurational space offered by the swap Monte Carlo algorithm [54–56], it now becomes feasible to study a broader range of system sizes over a broader range of temperatures and to carry out finite-size analyses to determine if the transitions persist in the thermodynamic limit.

In this paper, we present an extensive numerical study of the thermodynamics and phase transitions of constrained supercooled liquids in dimensions $d = 3$ and $d = 2$. We find strong signatures of the mean-field phenomenology for both cases when system sizes are sufficiently small. By using a careful finite-size scaling analysis, we show that the $3d$ phase diagram exhibits a first-order transition line ending in a RFIM-like critical point. A short report of this investigation on the critical behavior in $3d$ systems can be found in Ref. [57]. On the other hand, in $d = 2$, we find no signature of a phase transition down to the lowest temperature numerically accessible, which is below the extrapolated calorimetric glass transition temperature $T_g$. This is fully compatible with the RFIM phenomenology.

The rest of the manuscript is organized as follows. In Sec. II, we describe our numerical strategy. It relies on an optimized choice of a low temperature $T_0$ of the reference configurations, which is suggested by a mean-field analysis and is made possible by the swap algorithm. It is then combined with state-of-the-art importance sampling techniques. In Sec. III, we study the overlap statistics and the thermodynamics of constrained supercooled liquids in $2d$ and $3d$ for rather small samples. In Sec. IV, we perform finite-size analyses to capture the thermodynamic limit and determine the presence or absence of a transition in $3d$ and $2d$. In Sec. V, we focus on the $3d$ liquid and characterize the nature of the critical point at $(\epsilon_c, T_c)$. Finally, we summarize and discuss our results in

Sec. VI. Details on the mean-field analytical calculations are presented in an Appendix and those on the liquid models and the methods in another one.

## II. NUMERICAL STRATEGY

### A. Insights from the spherical $p$-spin model

So far, we have mostly discussed the $(\epsilon, T)$ phase diagram in the situation where the reference configurations are sampled at the same temperature as the constrained liquid, namely, $T = T_0$. In this case, the constrained liquid is attracted toward configurations which are typical of the unconstrained liquid at the same temperature $T$. One then has a handle on the organization of the typical metastable states at temperature $T$, in particular on their number which is controlled by the configurational entropy. However, one most generally has three control parameters, $\epsilon$, $T$, and $T_0$. Fixing the temperature $T_0$ of the reference configurations amounts to coupling the liquid at temperature $T$ to configurations which are typical at another temperature. The direct link to the configurational entropy is then lost because its contribution is intertwined with the intrinsic difference in the free energy of typical glassy states between $T$ and $T_0$. Yet, as we discuss below, interesting information can still be obtained while practical improvements are made possible.

The choice of $T_0$ affects the phase diagrams presented in Fig. 1(b)-(c). We have fully explored the influence of $T_0$ at the mean-field level. Detailed results are presented in Appendix A, and we merely summarize them here. We study the fully-connected spherical $p$-spin model (with $p \geq 3$) [58, 59]. Its Hamiltonian is given by

$$\widehat{H}_{\boldsymbol{J}}[\underline{\sigma}] = -\sum_{1 \leq i_1 < \cdots < i_p \leq N} J_{i_1 \ldots i_p} \sigma_{i_1} \ldots \sigma_{i_p}, \qquad (4)$$

where $\boldsymbol{J} = \{J_{i_1 \ldots i_p}\}_{1 \leq i_1 < \cdots < i_p \leq N}$ are Gaussian random variables of zero mean and variance $\mathbb{E}\{J_{i_1 \ldots i_p}^2\} = J^2 p!/(2N^{p-1})$, with $J > 0$, and the spin variables $\sigma_i$ are real numbers constrained to stay on the unit sphere. The model has already been extensively investigated (see, e.g., Ref. [60] for a review) and is known to exhibit a phenomenology similar to that of mean-field structural glasses [61, 62]. In particular, a random first-order transition (RFOT) at a nonzero temperature $T_K$ is found and the phase diagram in the $(\epsilon, T)$ plane is similar to that in Fig. 1(b) when $T = T_0$ [16, 17].

The thermodynamics of the spherical $p$-spin model can be computed exactly for any set of parameters $(\epsilon, T, T_0)$. We show in Fig. 2(a) the $(\epsilon, T)$ phase diagram for the case $T = T_0$ along with that for a low, fixed temperature $T_0$, focusing on the case $p = 3$. When $T = T_0$, a line of first-order transition emerges from the Kauzmann transition (RFOT) at $T_K$ and ends in a critical point at $(\epsilon_c^{(T=T_0)}, T_c^{(T=T_0)})$. When the temperature of the reference configurations is fixed to $T_0 < T_c^{(T=T_0)}$,

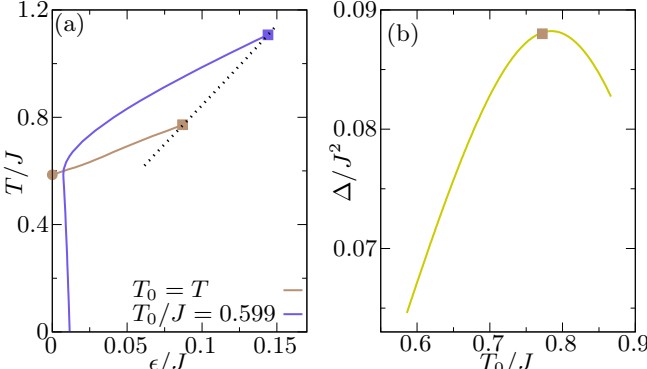

FIG. 2. (a) Phase diagram in the $(\epsilon, T)$ plane of the fully-connected spherical $p$-spin model ($p = 3$) for $T = T_0$ and for a fixed temperature $T_0$ of the reference configurations. Energies are expressed in units of the strength $J$ of the coupling constants between spins [see Eq. (4) and below]. When $T = T_0$, a first-order transition line emerges from $T_K/J = 0.586$ and ends in a critical point at $T_c^{(T=T_0)}/J = 0.772$ and $\epsilon_c^{(T=T_0)}/J = 0.087$ (the Boltzmann constant is set to unity). For $T_0 < T_c^{(T=T_0)}$, a critical point still exists at a higher temperature $T_c(T_0)$ and a larger value of the source $\epsilon_c(T_0)$. For this choice of $T_0$, the line of first-order transition terminates at ($\epsilon > 0, T = 0$). The dotted line represents the loci of the critical points when $T_0$ is varied. (b) Variance $\Delta$ of the effective random field in the mapping to the random-field Ising model (RFIM): see Appendix A. The maximum is obtained for $T_0/J \approx 0.784$, close to the case $T = T_0$ (square).

the critical point still exists, but its position is shifted to a higher temperature $T_c(T_0)$ and a larger value of the source $\epsilon_c(T_0)$. The behavior of the first-order transition line at low temperatures is however qualitatively different. For temperatures $T_0$ that are sufficiently low (in particular for $T_0 < T_d$, with $T_d$ the dynamical glass transition temperature), the transition line is reentrant and ends at a finite source $\epsilon$ in the limit of zero temperature. In Appendix A, we detail the possible shapes of the phase diagram $(\epsilon, T)$ as a function of $T_0$. We also compute the location of the critical point as a function of $T_0$, showing that $T_c$ and $\epsilon_c$ are decreasing functions of $T_0$: this is illustrated by the dotted line in Fig. 2(a).

We next consider the critical behavior of the $p$-spin model beyond mean-field, *i.e.*, by taking into account finite-dimensional fluctuations. In Ref. [22], it was shown that the critical point of the spherical $p$-spin is in the universality class of RFIM when $T = T_0$. In Appendix A, we extend this conclusion to any temperature $T_0$ of the reference configurations and we perform the explicit mapping. In particular, our computation provides the variance $\Delta$ of the effective random field that emerges in the mapping. It is displayed in Fig. 2(b) as a function of $T_0$. The effective strength of the random field decreases for both low and high values of $T_0$, the maximum being achieved for $T_0 \approx T_c^{(T=T_0)}$. As a consequence, the case $T = T_0$ corresponds to near maximal effective random-field dis-

order.

We can use these mean-field results to somehow optimize our numerical strategy. First, except very close to $\epsilon = 0$, the phase diagram is qualitatively unchanged when varying $T_0$ over a broad range. As we are primarily interested in assessing the existence of transitions in the $(\epsilon, T)$ plane, this implies that we can choose the most convenient value of $T_0$. As we have seen, the critical point and the first-order transition line are shifted upward in temperature when $T_0$ is low enough. Previous numerical works suggested that if the critical point survives in finite $d$ for $T = T_0$, it should be close to, or below the mode-coupling crossover [20, 24, 26, 40, 57]. Consequently, we can take advantage of the swap Monte Carlo algorithm to generate very stable equilibrium configurations at the lowest accessible temperatures $T_0$, close to or even below the extrapolated calorimetric glass transition temperature. This should allow us to shift all the relevant thermodynamic features to higher temperatures where equilibration is much easier. The potential downside is that the effective disorder is lower than in the case $T = T_0$, with the implication that the RFIM behavior could be more difficult to observe. (If disorder is too weak, the system near the critical point behaves up to some distance as the pure Ising model and RFIM physics only dominates beyond some crossover length [49] that could be quite large; as will be seen, this is not the case here.)

### B. Models and sampling methods

We use a hybrid algorithm [56] combining swap Monte Carlo moves and molecular dynamics to simulate the size-polydisperse system described in Ref. [55] and in Appendix B. Two particles $i$ and $j$ interact via the repulsive pairwise potential $v(r_{ij})/v_0 = (\sigma_{ij}/r_{ij})^{12} + v_c(r_{ij}/\sigma_{ij})$, where the function $v_c$ in the second term regularizes the potential, the force and its derivative at a cutoff distance $1.25\sigma_{ij}$, with $r_{ij}$ the relative distance and $\sigma_{ij}$ the cross-diameter. The distribution of particle diameters is $p(\sigma_i) \propto \sigma_i^{-3}$, and the interaction is nonadditive, $\sigma_{ij} = 0.5(\sigma_i + \sigma_j)(1 - \mu|\sigma_i - \sigma_j|)$, in order to maximize the glass-forming ability of this system and avoid fractionation and crystallization. The average diameter $\sigma$ of the particles is used as unit length ($\mu = 0.2$ in this unit), $v_0$ as unit temperature (the Boltzmann constant is set to unity) and $\sqrt{m\sigma^2/v_0}$ as unit time (with $m$ the mass of the particles). The model has been studied both in $d = 2$ [63–65] and $d = 3$ [40, 55, 65, 66], where characteristic temperature scales and dynamical properties have been determined.

Guided by the analysis of the mean-field $p$-spin model, we focus on equilibrium reference configurations at a low temperature $T_0 = 0.06 \gtrsim T_g(\approx 0.056)$ in $3d$ and $T_0 = 0.03 < T_g(\approx 0.068)$ in $2d$, which we generate with the help of the swap algorithm. To study the thermodynamics of the constrained liquid at a temperature $T$, we

do not impose a source $\epsilon$ because this direct approach suffers from several sampling issues. First, at high temperatures but close to the putative critical point, the dynamics (even with the swap Monte Carlo algorithm) slows down significantly [57]. This critical slowing down is due to the diverging thermodynamic fluctuations of the overlap. In random-field-like systems, the slowing down is far more spectacular than in pure systems as the relaxation time increases exponentially with the correlation length (instead of algebraically), a feature known as activated dynamic scaling [67, 68] (see also Ref. [57] and Sec. V). In addition, near the first-order transition line, sampling may be hindered due to large nucleation barriers between the metastable and stable phases.

We use instead state-of-the-art importance sampling techniques combining umbrella sampling [69–71] and subsequent histogram reweighting [72, 73], as described in Appendix B. All the simulations are run with $\epsilon = 0$. We add a biasing potential that forces the system to visit untypical values of the overlap which would otherwise never be sampled by using a direct approach. As a consequence, we are able to repeatedly visit very unlikely configurations and to cover the entire overlap range between 0 and 1. In such a two-step numerical strategy, we can compute for a given temperature $T$ and a given reference configuration $\boldsymbol{r}_0^N$ the probability distribution $\mathcal{P}_\epsilon(Q; \boldsymbol{r}_0^N)$ of the overlap for any source $\epsilon$ with a good numerical accuracy. From this distribution, the thermal average of any observable $\mathcal{A}(\widehat{Q})$ which only depends on the overlap can be computed as

$$
\begin{aligned}
\langle \mathcal{A}(\widehat{Q}) \rangle_\epsilon (T; \boldsymbol{r}_0^N) &= \frac{\int \mathrm{d}\boldsymbol{r}^N \mathcal{A}(\widehat{Q}[\boldsymbol{r}^N; \boldsymbol{r}_0^N]) e^{-\beta \widehat{H}_\epsilon [\boldsymbol{r}^N; \boldsymbol{r}_0^N]}}{\int \mathrm{d}\boldsymbol{r}^N e^{-\beta \widehat{H}_\epsilon [\boldsymbol{r}^N; \boldsymbol{r}_0^N]}} \\
&= \int_0^1 \mathrm{d}Q \mathcal{A}(Q) \mathcal{P}_\epsilon(Q; \boldsymbol{r}_0^N),
\end{aligned}
\tag{5}
$$

where $\beta = 1/T$. We then need to perform an average over the different realizations of the disorder, i.e., over the reference configurations,

$$
\begin{aligned}
\overline{\langle \mathcal{A}(\widehat{Q}) \rangle_\epsilon}(T, T_0) &= \frac{\int \mathrm{d}\boldsymbol{r}_0^N e^{-\beta_0 \widehat{H}[\boldsymbol{r}_0^N]} \langle \mathcal{A}(\widehat{Q}) \rangle_\epsilon (T; \boldsymbol{r}_0^N)}{\int \mathrm{d}\boldsymbol{r}_0^N e^{-\beta_0 \widehat{H}[\boldsymbol{r}_0^N]}} \\
&= \int_0^1 \mathrm{d}Q \mathcal{A}(Q) \overline{\mathcal{P}_\epsilon(Q; \boldsymbol{r}_0^N)}.
\end{aligned}
\tag{6}
$$

In particular, we will focus on the average overlap and on the amplitude of its fluctuations characterized by the overlap susceptibilities. As is usual for systems with quenched disorder, two susceptibilities can be defined to disentangle the two different sources (temperature and disorder) of fluctuations of the order parameter [74, 75].

The connected susceptibility

$$
\begin{aligned}
\chi_\epsilon^{(\mathrm{con})}(T, T_0) &= N\beta \left[ \overline{\langle \widehat{Q}^2 \rangle_\epsilon}(T, T_0) - \overline{\langle \widehat{Q} \rangle_\epsilon^2}(T, T_0) \right] \\
&= \overline{\chi_\epsilon(T; \boldsymbol{r}_0^N)},
\end{aligned}
\tag{7}
$$

where $\chi_\epsilon(T; \boldsymbol{r}_0^N) = N\beta[\langle \widehat{Q}^2 \rangle_\epsilon - \langle \widehat{Q} \rangle_\epsilon^2] = \partial \langle \widehat{Q} \rangle_\epsilon (T; \boldsymbol{r}_0^N) / \partial \epsilon$ is the thermal susceptibility for a fixed reference configuration accounting for the thermal fluctuations, and the disconnected susceptibility

$$
\chi_\epsilon^{(\mathrm{dis})}(T, T_0) = N\beta \left[ \overline{\langle \widehat{Q} \rangle_\epsilon^2}(T, T_0) - \overline{\langle \widehat{Q} \rangle_\epsilon}^2 (T, T_0) \right]
\tag{8}
$$

quantifies the fluctuations due to the disorder. The total susceptibility, computed as the second cumulant of the disorder-averaged probability distribution $\overline{\mathcal{P}_\epsilon(Q; \boldsymbol{r}_0^N)}$ of the overlap, is then given by

$$
\begin{aligned}
\chi_\epsilon^{(\mathrm{tot})}(T, T_0) &= N\beta \left[ \overline{\langle \widehat{Q}^2 \rangle_\epsilon}(T, T_0) - \overline{\langle \widehat{Q} \rangle_\epsilon}^2 (T, T_0) \right] \\
&= \chi_\epsilon^{(\mathrm{con})}(T, T_0) + \chi_\epsilon^{(\mathrm{dis})}(T, T_0),
\end{aligned}
\tag{9}
$$

which is simply the sum of the connected and disconnected contributions.

Before presenting our results, we comment on the potential difficulties stemming from the choice of the order parameter in the case of $2d$ systems. As the overlap $\widehat{Q}[\boldsymbol{r}^N; \boldsymbol{r}_0^N]$ is defined from the positions of the particles in Eq. (1), it may suffer in $d = 2$ from large collective translational displacements [76, 77] which have been associated with "Mermin-Wagner fluctuations" preventing periodic ordering in $2d$ systems. This would hamper the detection of the localized phase, irrespectively of the existence of a transition. Other choices for the order parameter (for instance the mean-squared displacement from the reference configuration [12] or the quadratic cumulative difference between the density fields in the constrained and reference replicas [78, 79]) suffer from the same issue. The amplitude of these fluctuations increases (linearly) with the temperature and (logarithmically) with the system size [80]. In consequence, for the system sizes and the temperatures that are considered here in $2d$ (up to $N = 250$), the Mermin-Wagner fluctuations are expected to be irrelevant. For instance, the translational (self-intermediate scattering function) correlation function and the bond-orientational correlation function $C_{\psi_6}(t)$ (see Appendix B) are very similar despite the fact that the former is sensitive to the Mermin-Wagner fluctuations and not the latter. For the system sizes and the temperatures considered, the relaxation times that are extracted from the two functions closely follow each other when varying the temperature [81].

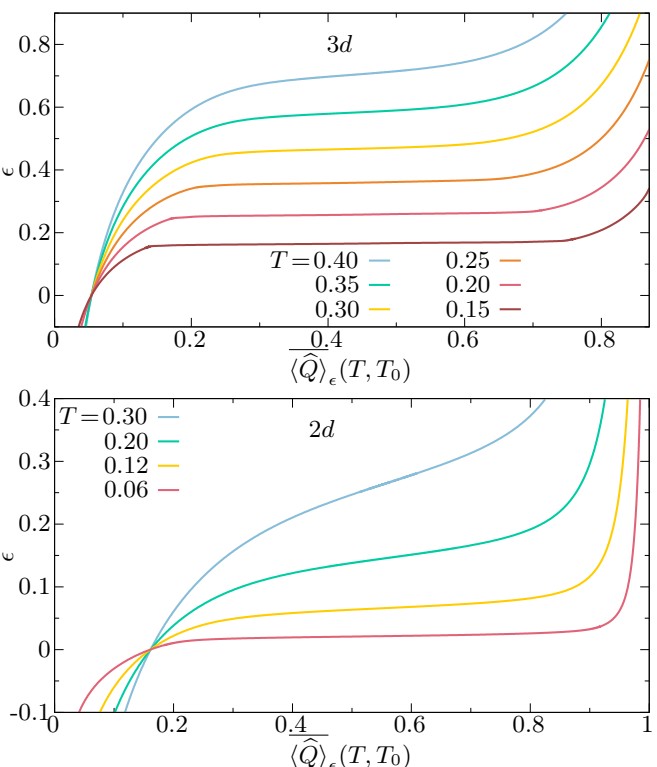

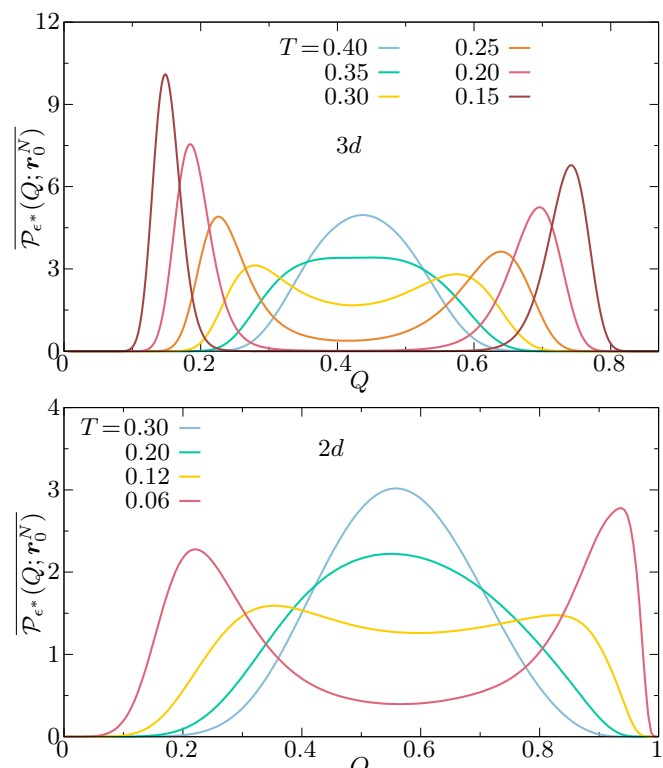

FIG. 3. Numerical isotherms showing the applied source $\epsilon$ versus the average overlap $\overline{\langle\widehat{Q}\rangle}_\epsilon(T, T_0)$ for several temperatures $T$ and a fixed temperature $T_0$ of the reference configurations. Top: $3d$ liquid with $N = 600$ and $T_0 = 0.06$. Bottom: $2d$ liquid with $N = 64$ and $T_0 = 0.03$. The isotherms are strictly monotonically increasing at high temperatures but become almost flat at low temperatures, as expected for a first-order transition ending in a critical point.

FIG. 4. Average probability distribution $\overline{\mathcal{P}_{\epsilon^*}(Q; \boldsymbol{r}_0^N)}$ of the overlap $Q$ for $\epsilon = \epsilon^*(T, T_0)$, at which the total variance of the overlap has a maximum, several temperatures $T$ and a fixed temperature $T_0$ of the reference configurations. Top: $3d$ liquid with $N = 600$ and $T_0 = 0.06$. Bottom: $2d$ liquid with $N = 64$ and $T_0 = 0.03$. With decreasing temperature, the probability distribution broadens and eventually becomes bimodal, which is a manifestation of a growing static lengthscale that exceeds the linear size of the system.

## III. MEAN-FIELD-LIKE BEHAVIOR IN FINITE SYSTEMS

### A. Thermodynamic properties in the presence of a source $\epsilon$ in $d = 3$ and $d = 2$

We first consider the thermodynamic properties of the constrained liquid when the source $\epsilon$ is applied on relatively small systems in $d = 3$ ($N = 600$) and $d = 2$ ($N = 64$). Isotherms are shown in Fig. 3. They correspond to the source $\epsilon$ plotted versus the average overlap order parameter $\overline{\langle\widehat{Q}\rangle}_\epsilon(T, T_0)$ for several temperatures $T$ at a fixed temperature $T_0$ of the reference configurations. The latter is chosen as $T_0 = 0.06$ in $3d$ and 0.03 in $2d$, and the definition of the double average is given in Eqs. (5)-(6). Imposing a finite positive (respectively, negative) $\epsilon$ biases the overlap toward larger (respectively, smaller) values than its "random" value $Q_{\rm rand}$. Isotherms are strictly monotonically increasing at large temperatures with an inflexion point that corresponds to maximal fluctuations at a fixed temperature $T$. Indeed, from Eq. (7), it is easy to see that the slope of the tangent to

the isotherm corresponds to the inverse of the connected susceptibility, susceptibility which then has a maximum at the inflexion point. As the temperature $T$ decreases, the value of $\epsilon$ beyond which the system is localized also decreases, as the attraction between configurations has to counterbalance smaller thermal fluctuations (or equivalently a smaller entropic cost). At the same time, the slope at the inflexion point of the isotherm decreases until the lowest temperatures at which the isotherms seem to plateau. This directly indicates growing fluctuations of the order parameter when decreasing the temperature.

This behavior is consistent with phase coexistence between low- and high-overlap phases at low temperatures ending in a critical point at a larger temperature. The curves in Fig. 3 are reminiscent of the van der Waals isotherms for the liquid-gas transition when corrected by the Maxwell construction [82]. The average overlap being here computed in the canonical ensemble in which $\epsilon$ is the control parameter, the isotherms cannot display any loop: the isotherms as calculated involve $\overline{\langle\widehat{Q}\rangle}_\epsilon(T, T_0)$ which is the first cumulant of the overlap distribution

and therefore takes a unique value at any given $\epsilon$ in a finite-size system. (Loops could be observed in a "micro-canonical" iso-overlap ensemble which, in the case of phase coexistence, is not equivalent to the canonical ensemble for a finite-size system.) Isotherms in the canonical ensemble can become strictly flat, but in the thermodynamic limit only. For finite-size systems, they display a residual slope of order $1/\sqrt{N}$ in disordered systems: see Eq. (12). A finite-size analysis is therefore necessary to detect whether the remnants of the mean-field phenomenology seen in relatively small systems persist as a true phase transition in the thermodynamic limit.

As mentioned in the previous section, our numerical strategy not only enables us to measure the average overlap but also its full probability distribution averaged over the reference configurations, $\overline{\mathcal{P}_\epsilon(Q;\boldsymbol{r}_0^N)}$, for any source $\epsilon$. From our discussion of the isotherms we know that, at a fixed temperature $T$, the connected susceptibility displays a maximum for some intermediate value of the source and that this maximum increases with decreasing temperature. Actually, both the connected and the disconnected susceptibilities are maximum around the same value of $\epsilon$, and we let $\epsilon^*(T,T_0)$ denote the value of the source at which the total susceptibility, which is the sum of the connected and disconnected susceptibilities [see Eq. (9)], is maximum. We then display in Fig. 4 the disorder-averaged probability distribution of the overlap for several temperatures $T$, a fixed temperature $T_0$ of the reference configurations, and $\epsilon = \epsilon^*(T,T_0)$. At high temperatures, the distribution is almost Gaussian with a single peak centered at $Q$ close to its average value. As the temperature $T$ decreases, the overall width of the distribution increases, reflecting larger overlap fluctuations as already inferred from the slope of the isotherms. Eventually, the distribution becomes strongly bimodal for the lowest temperatures. This is exactly what is expected if there is a phase separation between a delocalized and a localized phase, corresponding to a first-order transition line in the $(\epsilon,T)$ phase diagram.

One should of course be cautious before concluding to the existence of a phase transition, as this requires a finite-size analysis. Nonetheless, the fact that the probability distribution becomes increasingly bimodal for a given system size as one lowers the temperature is evidence for the existence of a static (thermodynamic) lengthscale associated with overlap fluctuations that grows with decreasing temperature. This is consistent with the existence of a critical point at a finite temperature $T_c$, at which the lengthscale would diverge. At this point however, several other scenarios cannot be excluded, such as a divergence at zero temperature only or a growth without divergence of the correlation length: see the schematic phase diagrams in Fig. 1(b)-(d).

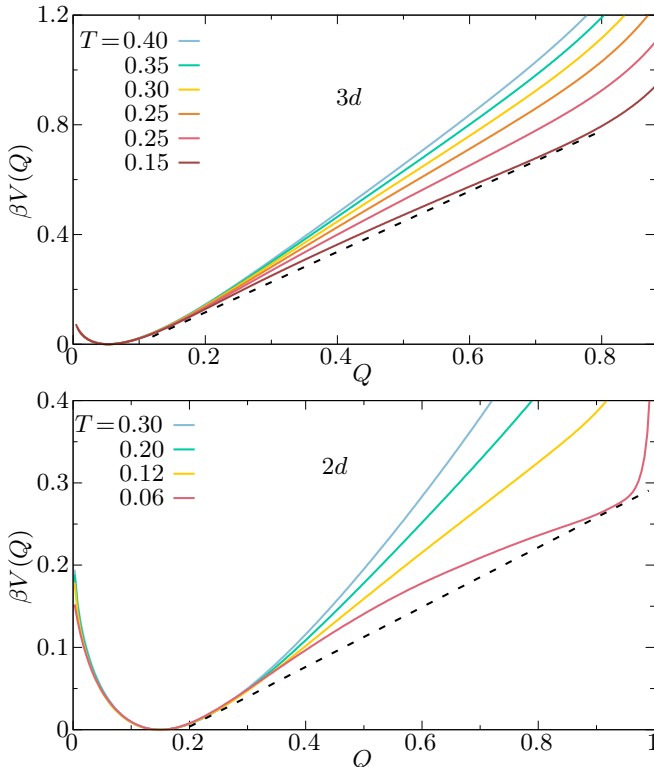

FIG. 5. Franz-Parisi (FP) potential rescaled by the temperature, $\beta V(Q)$, for the 3d liquid ($N = 600$, top) and the 2d liquid ($N = 64$, bottom) at several temperatures $T$ for a fixed temperature $T_0$ of the reference configurations ($T_0 = 0.06$ for $d = 3$ and $T_0 = 0.03$ for $d = 2$). The potential is convex at high temperatures with a single minimum at $Q = Q_{\mathrm{rand}}$. However, with the relatively small system sizes considered, it becomes nonconvex at lower temperatures (the dashed lines are a guide to the eye). This behavior is similar to that of mean-field glass-formers: compare with Fig. 12 in Appendix A.

## B. Evolution with the temperature of the Franz-Parisi potential

The Franz-Parisi (FP) potential is the free-energy cost for keeping equilibrium liquid configurations at a given value of the overlap with a reference configuration, chosen here at a fixed temperature $T_0$. It is defined as the large deviation rate function of the probability distribution of the overlap when $\epsilon = 0$, i.e., [15, 16]

$$V(Q) = -\frac{T}{N}\overline{\ln \mathcal{P}_{\epsilon=0}(Q;\boldsymbol{r}_0^N)} = \overline{V(Q;\boldsymbol{r}_0^N)}. \qquad (10)$$

The FP potential is defined up to an irrelevant additive constant, which we fix so that it vanishes at its absolute minimum.

We show in Fig. 5 the temperature evolution of the FP potential for a 3d system with $N = 600$, $T_0 = 0.06$ and a 2d one with $N = 64$, $T_0 = 0.03$. The trends are similar in both cases. The FP potential always displays an absolute minimum for $Q = Q_{\mathrm{rand}}$, reflecting the fact that in the

temperature range which we are able to simulate, the liquid is always found in the delocalized state when $\epsilon = 0$. The potential is strictly convex at high temperatures but becomes slightly nonconvex at the lowest temperatures (compare with the dashed lines). This behavior is reminiscent of that observed in mean-field glass-formers (see, *e.g.*, Fig. 12 for the fully connected spherical $p$-spin model in Appendix A). However, in the present situation, the nonconvexity results from a finite-size effect that limits the spatial extent of the fluctuations and is due to the rather small system sizes considered. Convexity needs to be restored in finite-dimensional systems in the thermodynamic limit ($N \to +\infty$) [83].

The thermodynamics of the constrained liquid, *i.e.*, the liquid in the presence of a nonzero applied source $\epsilon$, can be directly obtained from the FP potential, and this gives a complementary picture to that presented in the preceding subsection. For a given source $\epsilon$, it is convenient to tilt the FP potential according to $V_\epsilon(Q) = V(Q) - \epsilon Q$. The latter is related to the free energy as a function of the applied source $F(\epsilon)$ via a Legendre-Fenchel transform: $F(\epsilon) = \inf_Q\{V_\epsilon(Q)\} = V_\epsilon(Q_\epsilon^*(T, T_0))$ with $V_\epsilon'(Q_\epsilon^*(T, T_0)) = 0$, where a prime denotes a derivative with respect to the argument. At high temperatures, the FP potential is strictly convex and so is the tilted potential $V_\epsilon(Q)$. The FP potential can then be written as the Legendre-Fenchel transform of $F(\epsilon)$, namely $V(Q) = \sup_\epsilon\{F(\epsilon) + \epsilon Q\}$, resulting in $F'(\epsilon) = -Q_\epsilon^*(T, T_0) = -\langle \widehat{Q} \rangle_\epsilon(T, T_0)$. At the lowest temperatures shown in Fig. 5 for $d = 3$ and $d = 2$ the FP potential has lost convexity, which implies that for a range of values of $\epsilon$ the tilted potential $V_\epsilon(Q)$ is also nonconvex and has two minima and one maximum. For a specific value $\epsilon^*(T, T_0)$ the two minima have the same height, which corresponds in a mean-field setting to a first-order transition between a low-overlap and a high-overlap phase and in the present finite-size finite-dimensional systems to a vestige of such a transition [84]. In a finite-dimensional system in the thermodynamic limit, the FP potential must be convex but can nonetheless display a linear segment between two values $Q_{\text{low}}$ and $Q_{\text{high}}$ of the overlap. The slope of this segment is the source $\epsilon^*(T, T_0)$ at which phase coexistence between the low-overlap phase with $Q = Q_{\text{low}}$ and the high-overlap phase with $Q = Q_{\text{high}}$ takes place. The highest temperature at which this singular linear behavior exactly disappears then corresponds to the critical temperature $T_c$ and $V(Q)$ displays an inflexion point at the critical value $Q_{\text{low}} = Q_{\text{high}} = Q_c$ of the overlap. This corresponds to a critical source $\epsilon_c = \epsilon^*(T_c, T_0)$.

All of the above shows that glass-forming liquid models in $d = 3$ and $d = 2$ simulated with modest system sizes display a phenomenology similar to that of mean-field glass-formers. This is in line with the outcome of several previous simulation studies [17, 19, 20, 23, 24, 26, 27, 85, 86]. However, the presence of *bona fide* transitions in the $(\epsilon, T)$ diagram requires a finite-size study to determine whether the features seen in small systems persist when extrapolating to the thermodynamic limit.

## IV. FINITE-SIZE ANALYSIS: CONTRASTING $2d$ AND $3d$

### A. System-size dependence of the overlap probability distribution

To assess the existence of a first-order transition line ending in a critical point in the extended phase diagram of supercooled liquids in $2d$ and $3d$, we first analyze the system-size dependence of the probability distribution $\overline{\mathcal{P}_{\epsilon^*}(Q; \boldsymbol{r}_0^N)}$ of the overlap for two different temperatures: see Fig. 6.

At the lower temperature in $3d$ ($T = 0.15$), the probability distribution of the overlap is bimodal for all studied system sizes, with two maxima at $Q = Q_{\text{low}}$ and $Q = Q_{\text{high}}$. In addition, the distribution gets increasingly bimodal when the system size is increased: the width of the two peaks shrinks while the free-energy barrier between the two maxima,

$$\beta \Delta \mathcal{F}(T, T_0) = \ln\left[\frac{\sqrt{\overline{\mathcal{P}_{\epsilon^*}(Q_{\text{low}}; \boldsymbol{r}_0^N)}\,\overline{\mathcal{P}_{\epsilon^*}(Q_{\text{high}}, \boldsymbol{r}_0^N)}}}{\overline{\mathcal{P}_{\epsilon^*}(Q_{\text{min}}; \boldsymbol{r}_0^N)}}\right],$$
(11)

where $Q_{\text{min}}$ is the location of the relative minimum of the probability distribution in the range $[Q_{\text{low}}, Q_{\text{high}}]$, grows. The probability distribution then appears to converge to a double Dirac distribution in the thermodynamic limit. At the higher temperature in $3d$ ($T = 0.30$), the probability distribution is bimodal in small-enough samples ($N \lesssim 1000$) but this behavior disappears when considering large enough systems: see the curve for $N = 2400$. For this temperature, the distribution is therefore expected to become Gaussian in the thermodynamic limit. This pattern as a function of system size and temperature provides support to the existence of a critical point at a nonzero temperature $T_c \in [0.15, 0.30]$. We stress that the system sizes considered here are unprecedentedly large compared to earlier simulation studies of glass-forming liquids which were limited to at most a few hundreds of particles. Clearly, dealing with too small system sizes tends to overestimate the critical temperature $T_c$ and may even lead to an erroneous conclusion concerning the existence of a transition.

Consider now the case $d = 2$. The overlap probability distribution is bimodal in sufficiently small systems but its overall width always narrows and the distribution eventually becomes single-peaked in larger samples. Excluding the unlikely scenario in which bimodality reappears at even larger system sizes, this observation rules out the existence of a critical point in $2d$ for $T \geq 0.06$. We emphasize that with the help of the swap Monte Carlo algorithm, we have been able to prepare equilibrium configurations at $T_0 = 0.03$, *i.e.*, much lower than the estimated calorimetric glass transition temperature $T_g = 0.068$: they represent equilibrium reference configurations with an estimated (but unmeasurable!) relaxation time of about $10^{37}$ in the units of the model.

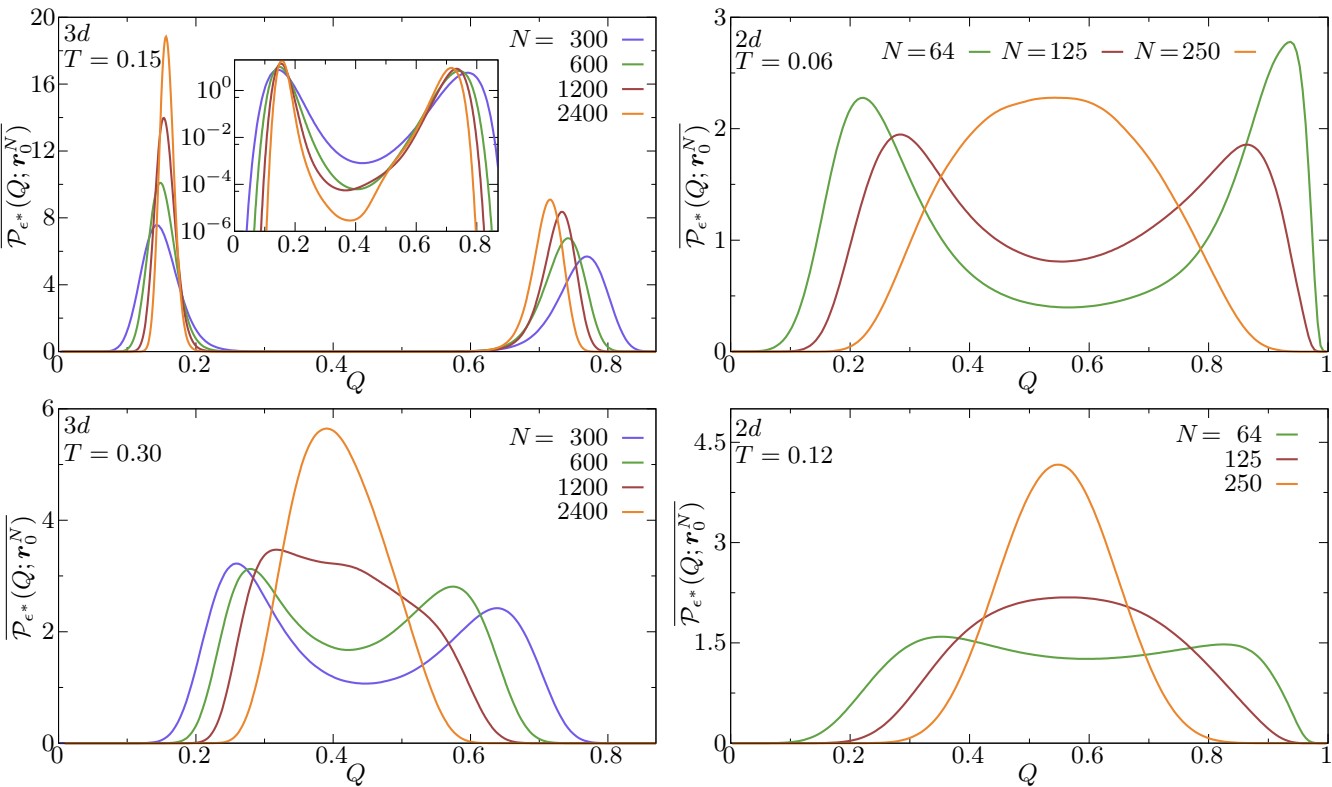

FIG. 6. Evolution with system size $N$ of the disorder-averaged probability distribution $\overline{\mathcal{P}_{\epsilon^*}(Q; r_0^N)}$ of the overlap $Q$ in $3d$ (left) and $2d$ (right) for $\epsilon = \epsilon^*(T, T_0)$, at which the total variance of the overlap order parameter is maximum. The distributions are shown for two different temperatures $T$ and a fixed temperature $T_0$ of the reference configurations ($T_0 = 0.06$ in $3d$ and $T_0 = 0.03$ in $2d$). The $3d$ results support the existence of a first-order transition line ending in a critical point at a temperature $0.15 \leq T_c < 0.30$. Instead in $2d$, the results point to the absence of a transition at any temperature $T \geq 0.06$. The inset in the top left panel shows the probability distributions in a logarithmic scale to highlight that the free-energy barrier between the low-overlap and high-overlap phases grows with system size.

Converted into physical units [65], this corresponds to about $10^{18}$ years, much larger than the age of the universe. In addition, the lowest temperature $T$ that we could achieve ($T = 0.06$) is itself below the extrapolated glass transition temperature $T_g$. This suggests the absence of phase transition in $2d$ and, to the least, we can conclude that in the experimentally relevant temperature range (near and above the calorimetric glass transition temperature), there is no signature of a critical point in the $2d$ glass-forming liquid. The fact that one needs to consider larger system sizes to recover a single-peaked probability distribution of the overlap as one lowers the temperature ($N \geq 125$ for $T = 0.12$ and $N \geq 250$ for $T = 0.06$) nonetheless indicates the existence of a growing static lengthscale associated with overlap fluctuations. Although we do not attempt to characterize its precise behavior due to the limited system sizes that we can access, our findings are compatible with the existence of a zero-temperature critical point in $d = 2$.

## B. Finite-size scaling in $3d$ indicates a first-order transition in the thermodynamic limit

To further confirm that the $3d$ constrained liquid is below a critical point when $T = 0.15$ and then undergoes a first-order transition as a function of the applied source $\epsilon$, we assess the validity of the scaling laws predicted by the mapping onto an effective random-field Ising model [21, 22]. We first display in Fig. 7 the system-size dependence of the connected and the disconnected susceptibilities evaluated at or very near their maximum, when $\epsilon = \epsilon^*(T, T_0)$. At a first-order transition in the presence of a random field, the finite-size scaling behavior is described by [74, 75]

$$\chi_{\epsilon^*}^{(\mathrm{con})}(T, T_0) \sim L^{d/2} \sim \sqrt{N},$$
$$\chi_{\epsilon^*}^{(\mathrm{dis})}(T, T_0) \sim L^d \sim N, \tag{12}$$

where $L \propto N^{1/d}$ is the linear extent of the system. The fingerprint of the random field is the dominance at large scale of the sample-to-sample fluctuations encoded in the disconnected susceptibility over the thermal ones encoded in the connected susceptibility [87]. As can be

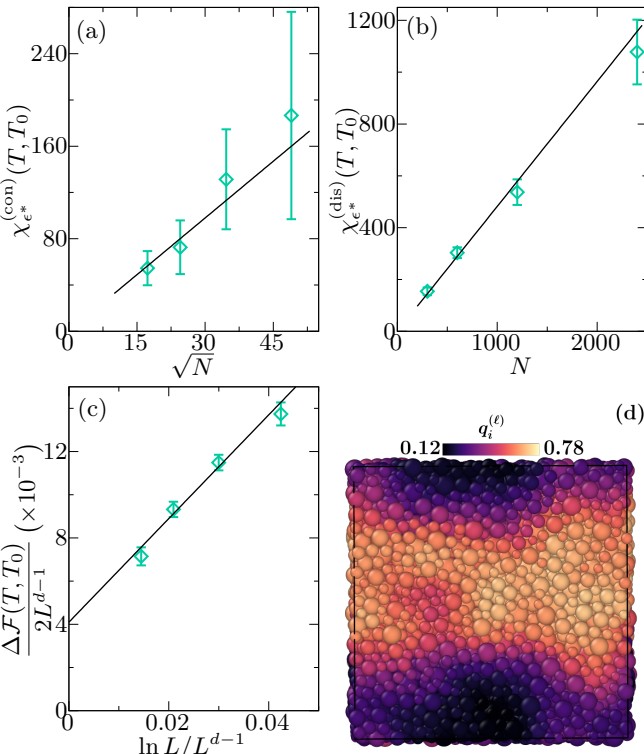

FIG. 7. Finite-size scaling analysis for the $3d$ constrained glass-forming liquid in the region of first-order transition. Maximum of (a) the connected $\chi_{\epsilon^*}^{(\mathrm{con})}(T, T_0)$ and (b) disconnected $\chi_{\epsilon^*}^{(\mathrm{dis})}(T, T_0)$ susceptibilities at $T = 0.15$ for a fixed temperature $T_0 = 0.06$ of the reference configurations. The full lines are the result of a linear fit to the data and are compatible with what is expected for the random-field Ising model [compare with Eq. (12)]. (c) Rescaled free-energy barrier $\Delta\mathcal{F}(T, T_0)/(2L^{d-1})$ versus $\ln L/L^{d-1}$ with $L \propto N^{1/d}$ the linear size of the system, validating the scaling in Eq. (13). The intercept corresponds to the surface tension between the low-overlap and the high-overlap phases, $\Upsilon(T, T_0) \approx 0.0041$. For panels (a)-(c) error bars are obtained from the jackknife method when performing the disorder average [73]. (d) Snapshot of the liquid with $N = 10000$ for $T = 0.15$, $T_0 = 0.06$ and a fixed value of the overlap with the reference configuration, $\widehat{Q} \approx 0.44$, intermediate between low and high overlaps. The particles are colored according to their coarse-grained overlap $q_i^{(\ell)}$ with the reference configuration: see Eqs. (14)-(15). A macroscopic phase separation is clearly visible.

seen from Fig. 7(a)-(b), both relations are well satisfied by our data, even though error bars are quite large for the largest system size.

We have also studied the size-dependence of $\Delta\mathcal{F}(T, T_0)$, the free-energy barrier separating the low-overlap and the high-overlap phases [see Eq. (11)]. If one assumes a planar interface between the two coexisting phases, the free-energy barrier should scale as [88]

$$\frac{\Delta\mathcal{F}(T, T_0)}{2L^{d-1}} = \Upsilon(T, T_0) + A\frac{\ln L}{L^{d-1}} + \frac{B}{L^{d-1}}. \quad (13)$$

In this equation, $A$ and $B$ are unknown coefficients char-

acterizing the amplitude of the subdominant behaviors while the factor of 2 in the denominator of the left-hand side comes from using periodic boundary conditions. The free-energy barrier per unit area converges to the surface tension $\Upsilon(T, T_0)$ when $L \to +\infty$.

To describe the subdominant terms we have added to the standard contribution proportional to $B$ an extra $\ln L/L^{d-1}$ dependence accounting for massless modes due to the invariance of the free-energy cost under translations of the planar interface and contributions from nonplanar interfaces [89]. For large-enough sizes (as is the case here), the latter contribution dominates the former one, and in Fig. 7(c) we show that the variation of $\Delta\mathcal{F}(T, T_0)/(2L^{d-1})$ is indeed consistent with a linear behavior as a function of $\ln L/L^{d-1}$. From the fit we extract a surface tension $\Upsilon(T, T_0) \approx 0.0041$ for $T = 0.15$. This positive nonzero value guarantees the self-consistency of our ansatz and confirms the presence of a phase separation associated with the first-order transition.

A snapshot of a configuration of the $3d$ constrained liquid with $N = 10000$, $T = 0.15$ and for a fixed temperature $T_0 = 0.06$ of the reference configuration is shown in Fig. 7(d). This configuration is obtained during a biased simulation with an umbrella potential chosen so that the overlap with the reference configuration is intermediate between $Q_{\mathrm{low}}$ and $Q_{\mathrm{high}}$: $\widehat{Q} \approx 0.44$; macroscopic phase separation is then expected. For each particle, we compute a local overlap

$$q_i = \sum_{j=1}^{N} w(|\boldsymbol{r}_i - \boldsymbol{r}_j^{(0)}|/a), \quad (14)$$

where the sum runs over all the particles of the reference configuration and $w(x)$ is the window function already introduced in Eq. (1). To smooth out the local fluctuations of the overlap we coarse-grain this single-particle quantity by using an exponential window of size $\ell = 1$, which leads to

$$q_i^{(\ell)} = \frac{\sum_j q_j e^{-r_{ij}/\ell}}{\sum_j e^{-r_{ij}/\ell}}, \quad (15)$$

where the sums run over all the particles in the constrained replica, and $r_{ij} = |\boldsymbol{r}_i - \boldsymbol{r}_j|$. We clearly observe that the system segregates into two phases with distinct values of the overlap. The interface is not perfectly planar and there are inhomogeneities of the overlap inside the high-overlap phase. Nonetheless, all the particles with a local overlap larger than the average form a single connected cluster: their relative distance is smaller than 1.5, which corresponds to the first minimum in the radial pair correlation function $g(r)$ [90]. This snapshot illustrates what a phase separation in a constrained glass-forming liquid looks like, and it strengthens the conclusions of the scaling analysis of the free-energy barrier following Eq. (13).

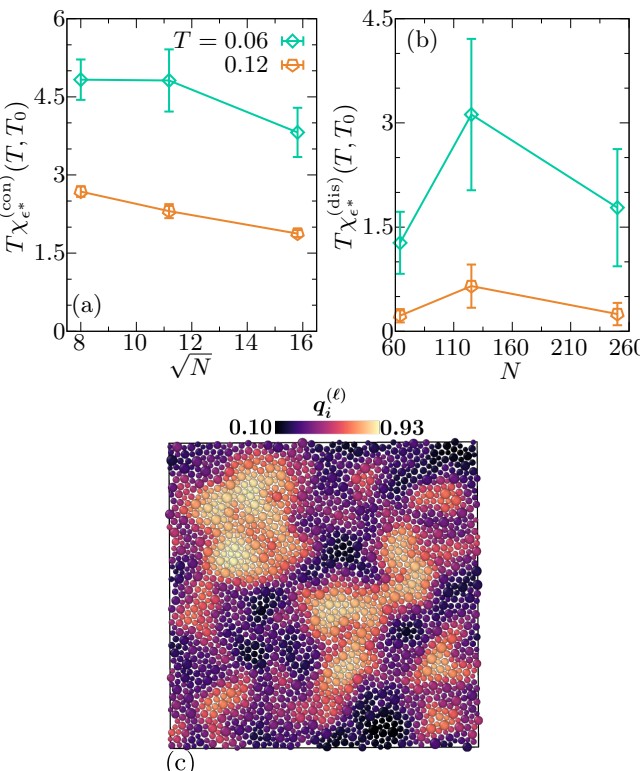

FIG. 8. Finite-size analysis for the $2d$ constrained glass-forming liquid. (a) Maximum of the connected susceptibility $T\chi_{\epsilon^*}^{(\mathrm{con})}(T,T_0)$ as a function of the linear system size $\sqrt{N}$ and (b) maximum of the disconnected susceptibility $T\chi_{\epsilon^*}^{(\mathrm{dis})}(T,T_0)$ as a function of the system size $N$ for two temperatures $T = 0.06$ and $T = 0.12$ and a fixed temperature $T_0 = 0.03$ of the reference configurations. Contrary to the $3d$ liquid, the susceptibilities are bounded, do not grow with system size and should then remain finite in the thermodynamic limit. Error bars are obtained from the jackknife method when performing the disorder average. (c) Snapshot of the liquid with $N = 2000$, $T = 0.06$, and a fixed value of the overlap with the reference configuration, $\widehat{Q} \approx 0.48$, which is intermediate between low and high overlaps. The particles are colored according to their coarse-grained overlap $q_i^{(\ell)}$ with the reference configuration: see Eqs. (14)-(15). Contrary to the $3d$ liquid, no macroscopic phase separation is observed.

## C. Finite-size analysis in $2d$ shows no sign of phase transition

We give further support to the absence of a phase transition in $2d$ in the thermodynamic limit for the whole accessible temperature range. We plot in Fig. 8(a,b) the maximum of the connected and the disconnected susceptibilities for two temperatures $T = 0.12$ and $T = 0.06$ and a fixed temperature of the reference configurations, $T_0 = 0.03$. We observe that, contrary to what is found for the $3d$ system, the susceptibilities do not grow with system size and therefore will most likely converge to a finite value in the thermodynamic limit. (Of course, with only so few points we did not try to perform any *bona fide* scal-

ing analysis [91].) Accordingly, in real space, the system does not phase separate: this is illustrated in Fig. 8(c) which is obtained in the course of an umbrella sampling simulation at $T = 0.06$ for a larger system of $N = 2000$ particles. Instead of a system-spanning phase separation, the $2d$ liquid constrained at an intermediate value of the overlap with the reference configuration displays small domains characterized by either a small or a large overlap, and the particles with an overlap larger than the average one do not form a single connected cluster. This is in contrast with the macroscopic phase separation observed in $d = 3$ and corroborates the conclusion drawn above from the system-size dependence of the overlap probability distribution.

## D. Further results concerning $2d$ and $3d$ for the case with $T = T_0$

To confirm the conclusions obtained for a fixed low $T_0$ we have also studied the phase diagram of the $3d$ and $2d$ liquids in the case where the constrained liquid configurations and the reference configurations are at the same temperature, $T_0 = T$. This situation more directly probes the relevant regions of the underlying landscape and the physics of the glass-forming liquid in the absence of an applied source than when $T_0$ is fixed because the reference configurations are then typical states. However, as already stressed, such a study with $T_0 = T$ is computationally more demanding: if present, the critical point is indeed expected at a temperature $T_c$ at which the relaxation time of the unconstrained liquid is already so large that conventional simulation techniques without swap moves are barely able to equilibrate the system. In consequence, we have only probed the existence of a transition in $d = 3$ and the absence of a transition in $d = 2$ without delving more into the details.

We focus on the behavior of the disorder-averaged overlap probability distribution $\overline{\mathcal{P}_{\epsilon^*}(Q; \boldsymbol{r}_0^N)}$ for $\epsilon = \epsilon^*(T)$ (where the total variance of the overlap order parameter is maximum) which, as illustrated above, is a convenient means to contrast $2d$ and $3d$ physics. Both for $d = 2$ and $d = 3$ we display two temperatures and we study three and two system sizes respectively: see Fig. 9. When comparing with Fig. 6 obtained for a fixed low temperature $T_0$, one can see that one must go to significantly lower temperatures $T$ to observe a bimodal distribution of the overlap even for the smaller system sizes ($N = 300$ in $3d$ and $N = 64$ in $2d$): this illustrates the already emphasized trend with $T_0$ (see, *e.g.*, Sec. II). The $3d$ results, which have already been displayed in our short report [57], point to the persistence of a phase transition in the thermodynamic limit. For the lowest temperature considered, the two peaks at low and high overlap indeed grow and narrow as the system size increases, which suggests that the system is below the critical temperature, contrary to what is observed at the higher temperature.

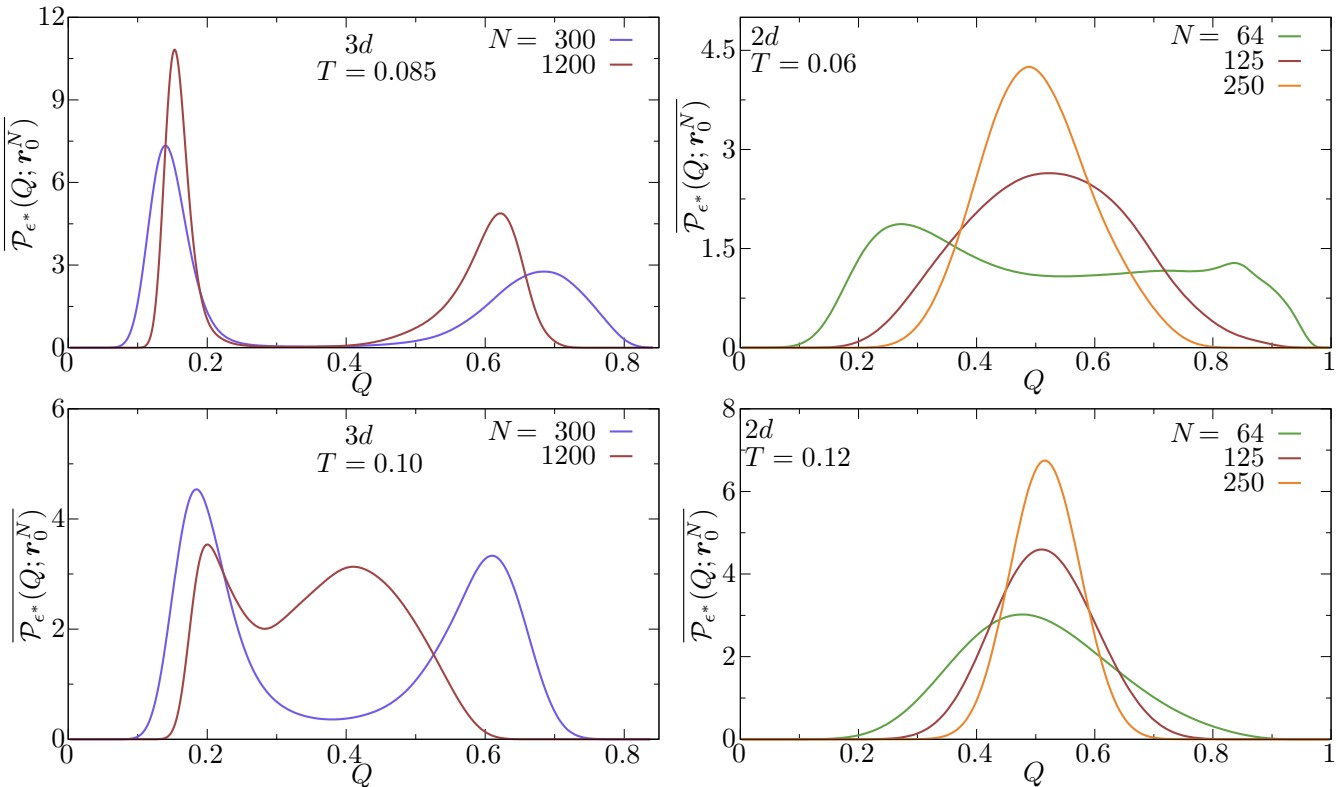

FIG. 9. Evolution with system size $N$ of the disorder-averaged probability distribution $\overline{\mathcal{P}_{\epsilon^*}(Q; \boldsymbol{r}_0^N)}$ of the overlap $Q$ in $3d$ (left) and $2d$ (right) for $\epsilon = \epsilon^*(T)$, at which the total variance of the overlap order parameter is maximum. The distributions are shown for two and three different temperatures $T$ respectively and the reference configurations are sampled at a temperature $T_0 = T$. The $3d$ results support the existence of a critical point at a temperature $0.085 \leq T_c < 0.100$. Instead in $2d$, the results point to the absence of a transition at any temperature $T \geq 0.06$ (which is below the estimated calorimetric glass transition temperature $T_g \approx 0.068$).

In $d = 2$ instead, our new results confirm the absence of a transition in the experimentally relevant temperature regime (the lowest temperature shown in the figure is below the calorimetric glass transition temperature $T_g$): the bimodal behavior of the overlap distribution, if present in small systems, disappears for a large enough size, which is sufficient to rule out the presence of a transition at these temperatures.

## V. CHARACTERIZATION OF THE CRITICAL POINT IN $3d$

In order to locate and characterize the critical point in the $3d$ constrained liquid, we focus on the analysis of the finite-size behavior of the connected and disconnected susceptibilities. (As discussed in Ref. [57] the conventional way of detecting a critical point through ratios of cumulants of the order parameter is not practical in the present case of a random-field-like system without $Z_2$ inversion symmetry.) When approaching close enough to a critical point in a finite-size system, the correlation length saturates around the linear size $L$ of the system. As a result, when considered at $\epsilon = \epsilon^*(T, T_0)$ (above the critical point this is the Widom line), the susceptibilities should

follow finite-size scaling relations [74],

$$
\begin{aligned}
\chi_{\epsilon^*}^{(\mathrm{con})}(T, T_0) &= L^{2-\eta} \widetilde{\chi}_{\mathrm{con}}(t L^{1/\nu}), \\
\chi_{\epsilon^*}^{(\mathrm{dis})}(T, T_0) &= L^{4-\overline{\eta}} \widetilde{\chi}_{\mathrm{dis}}(t L^{1/\nu}),
\end{aligned}
\tag{16}
$$

where $\widetilde{\chi}_{\mathrm{con}}$ and $\widetilde{\chi}_{\mathrm{dis}}$ are (non-singular) scaling functions, $\eta$, $\overline{\eta}$ and $\nu$ are critical exponents, and $t = T/T_c - 1$ is the reduced temperature. We expect the critical point to belong to the universality class of the random-field Ising model (RFIM) and we therefore take the values that have been accurately measured in the RFIM at zero temperature [92–94]: $\eta \approx 0.52$, $\overline{\eta} \approx 1.04$, and $\nu \approx 1.37$ (limiting ourselves here to two significant figures). One may note that $\overline{\eta} \approx 2\eta$. Although the relation is only approximate [93, 95], the deviations are very small in $3d$ and beyond the precision needed here. Then, combining Eqs. (12), (16), and the approximate relation between $\overline{\eta}$ and $\eta$, one obtains that the disconnected susceptibility scales as the square of the connected one both for the first-order transition region and near the critical point. More precisely,

$$
\chi_{\epsilon^*}^{(\mathrm{dis})}(T, T_0) \approx \frac{\Delta}{T_c} \chi_{\epsilon^*}^{(\mathrm{con})}(T, T_0)^2,
\tag{17}
$$

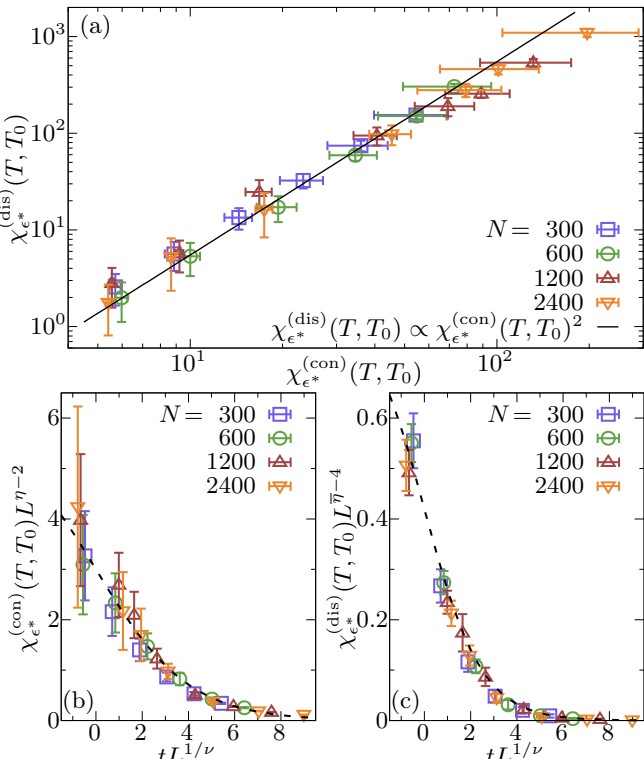

FIG. 10. Finite-size scaling analysis of the connected and disconnected susceptibilities in the $3d$ liquid close to the critical point for a fixed temperature $T_0 = 0.06$ of the reference configurations. (a) Scatter plot of the maximum value of the disconnected susceptibility $\chi_{\epsilon^*}^{(\text{dis})}(T, T_0)$ versus the maximum value of the connected susceptibility $\chi_{\epsilon^*}^{(\text{con})}(T, T_0)$. The full line represents the quadratic relation characteristic of the $3d$ random-field Ising model (RFIM). (b) Rescaled connected susceptibility and (c) rescaled disconnected susceptibility versus rescaled reduced temperature $t = T/T_c - 1$. $L \propto N^{1/3}$ is the linear system size. With the critical exponents taken as those of the $3d$ RFIM, a good data collapse is obtained for $T_c \approx 0.17$. The dashed lines are a guide for the eye. All error bars are obtained from the jackknife method when performing the disorder average.

where $\Delta$ represents the variance of the effective random field that emerges in the mapping from the constrained supercooled liquid to the RFIM while $T_c$ is the critical temperature. The dominance of the sample-to-sample fluctuations characterized by the disconnected susceptibility stems from the property that the critical behavior of the RFIM is controlled in a renormalization-group sense by a zero-temperature fixed point [96]. In Fig. 10(a), we show the scatter plot of the maximum of the disconnected susceptibility versus that of the connected susceptibility for a fixed temperature $T_0 = 0.06$ of the reference configurations. The above relation is well satisfied by our data. The disconnected susceptibility is larger than the connected one at low-enough temperatures or large-enough system sizes, which means that quenched disorder is relevant for the system. This is a

first evidence of random-field-like physics in the transition from the delocalized state to the localized state.

We now turn to the direct finite-size scaling analysis of the two susceptibilities by means of Eq. (16). In Fig. 10(b)-(c), we show the collapse of the properly rescaled connected and disconnected susceptibilities as a function of the reduced temperature. The critical temperature $T_c$ entering in the reduced temperature is the unique adjustable parameter to ensure the best data collapse on a master curve. (As mentioned above, the critical exponents are fixed to their known values: we did not try to fit the critical exponents from our data to reduce the number of free parameters.) Even though mixing-field effects may be present [97, 98], we find that a good collapse is obtained for $T_c \approx 0.17$. This estimate of the critical temperature is found by minimizing the average quadratic difference between the rescaled data and an *a priori* unknown master curve [99] by using the algorithm given in Refs. [100, 101].

All of the above confirms the existence in the $3d$ constrained liquid of a critical point in the universality class of the RFIM at a finite temperature $T_c$ and a finite applied source $\epsilon_c$ [with $\epsilon_c = \epsilon^*(T_c, T_0) \approx 0.20$], in agreement with field-theoretical treatments [21, 22]. The fact that no such critical point was detected in $d = 2$ is also fully in line with the mapping to the RFIM. The lower critical dimension of the latter is indeed $d = 2$ [49, 50, 52], so that $T_c$ should go to 0 for two-dimensional glasses.

From the prefactor obtained by fitting Eq. (17) and by using our estimate of the critical temperature $T_c$, we obtain an estimate of the strength of the effective disorder in the $3d$ liquid, $\sqrt{\Delta} \approx 0.097$. In the $3d$ RFIM one knows from numerical simulations [92, 93] that the disorder destroys the transition whenever $\sqrt{\Delta}/\mathcal{J} \gtrsim 2.3$, where $\mathcal{J}$ is the magnitude of the (ferromagnetic) coupling between the Ising spins. Accessing the value of this ratio in the $3d$ liquid would therefore provide an interesting consistency check for the existence of the transition. Unfortunately, although the effective coupling constant $\mathcal{J}$ may in principle be estimated from the surface tension $\Upsilon(T, T_0)$, the latter must be computed at temperatures significantly below $T_c$, because the surface tension vanishes at the critical point [68]. More specifically at the RFIM critical point, the free-energy barrier $\Delta\mathcal{F}$ crosses over from a dependence in $L^{d-1} \sim L^2$ to one in $L^\theta$ with $\theta = 2 + \eta - \bar{\eta} \approx 1.49$ the temperature exponent: see the inset in Fig. 2(a) in our previous paper [57]. Such an investigation at low temperatures is presently out of reach to computer simulations of constrained glass-forming liquids.

Finally, for completeness, we recall the results already given in our previous paper concerning the critical slowing down of the $3d$ constrained liquid near the critical point [57]. In the case of the RFIM, the time $\tau$ for relaxation to equilibrium diverges at the critical point but it does so in an anomalous manner. Instead of the conventional power-law behavior between the time and the correlation length, $\tau \sim \xi^z$ [102], one finds a much stronger

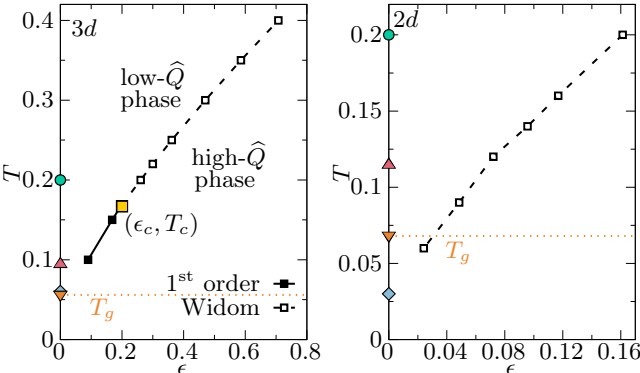

FIG. 11. Phase diagram of the glass-forming liquid in the $(\epsilon, T)$ plane in $d = 3$ (left) and $d = 2$ (right) for a fixed temperature of the reference configurations ($T_0 = 0.06$ in $3d$ and $T_0 = 0.03$ in $2d$). We show the loci of the maxima of total susceptibility, i.e., $\epsilon^*(T, T_0)$. In $3d$, a critical point (full yellow square) at $T_c \approx 0.17$ and $\epsilon_c \approx 0.20$ terminates the line of first-order transition (full line) and above it a Widom line is displayed as a dashed line. In $2d$, there is no critical point nor first-order transition line and only remains a Widom line (dashed line). In both panels, we give several characteristic temperatures: the onset temperature of glassy behavior (green disk), the mode-coupling crossover temperature (pink up triangle), the extrapolated calorimetric glass transition temperature $T_g$ (orange down triangle and horizontal dotted line), and the temperature $T_0$ of the reference configurations (blue diamond).

divergence, $\ln \tau \sim \xi^\psi$ with $\psi > 0$ a new exponent which in $3d$ is predicted to be equal to the temperature exponent $\theta \approx 1.49$ [103]. Furthermore, the time-dependent correlation function of the order parameter at long times is not as usual a function of $t/\tau$ but rather of $\ln t/\ln \tau$. These features, which are referred to as activated dynamic scaling, stem from the fact that the critical point is controlled by a zero-temperature fixed point [67, 68]. We have computed the equilibrium time-dependent correlation function of the fluctuations of the overlap in the $3d$ constrained liquid in the vicinity of the previously located critical point at $(\epsilon_c, T_c)$ and we have found that both predictions of activated dynamic scaling are obeyed by our data: see Ref. [57] for more details. This provides additional evidence that criticality in constrained glass-forming liquids is in the same universality class as the one of the RFIM.

## VI. SUMMARY AND DISCUSSION

Focusing on the insight that can be obtained about $3d$ and $2d$ glass-forming liquids from studying the statistical mechanics of the overlap between equilibrium and reference configurations, we have found two sets of results. First, we have confirmed that the mean-field scenario of glass formation which is based on the emergence of a complex free-energy landscape comprising a multi-

tude of metastable states is relevant to describe systems of relatively small sizes in which the spatial extent of the fluctuations (here, of the overlap order parameter) are by construction limited. Second, we have been able to simulate much larger system sizes than previously done on model supercooled liquids and thereby to carry out finite-size analyses in $d = 3$ and $d = 2$.

Our findings from extensive investigations of the phase diagrams of $3d$ and $2d$ liquids in the presence of an additional control parameter $\epsilon$ that introduces a bias toward high overlap with the reference configurations are summarized in Fig. 11. The results are displayed for low values of the temperature $T_0$ of the reference configurations, which are about (in $3d$) or much below (in $2d$) the extrapolated calorimetric glass transition temperature $T_g$. We give evidence that the mean-field prediction of a line of first-order transition between a low-overlap (delocalized) phase and a high-overlap (localized) phase terminating at a critical point persists in the thermodynamic limit in the $3d$ liquid but is absent in the $2d$ one at least down to temperatures that go below the calorimetric glass transition temperature $T_g$. In the $2d$ case, one still observes the analog of a Widom line with a growing correlation length as the temperature decreases but no sign of a critical point, and hence of a transition, in the accessible region of temperature. Although we have not carried out a similarly extensive investigation for the case where the constrained liquid and the reference configurations are at the same temperature, i.e., $T = T_0$, because it is computationally much more demanding, our results show the same pattern concerning $3d$ and $2d$ liquids. These observations, together with the results of a finite-size scaling analysis and a study of the relaxation dynamics near the critical point for the $3d$ liquid, are consistent with the prediction that the critical behavior terminating the transition between low- and high-overlap phases is in the universality class of the random-field Ising model.

Our conclusions are compatible with previous studies on the same model glass-forming liquids in which measurements of the configurational entropy were performed [38, 39, 63]. The outcome of these studies is that whereas the $3d$ curve showing the temperature dependence of the configurational entropy seems to extrapolate to a vanishing value at a nonzero $T_K$, the extrapolation of the $2d$ curve instead points to $T_K = 0$. The entropy crisis at $T_K$ being the endpoint at $\epsilon = 0$ of the first-order transition line in the $(\epsilon, T)$ diagram when $T_0 = T$ and the critical point at $(\epsilon_c, T_c)$ being the upper limit of the line, $T_K \neq 0$ requires $T_c \neq 0$ and, on the other hand, $T_c = 0$ implies $T_K = 0$ (or no $T_K$ at all). With the additional property that $T_c^{(T=T_0)}$ is less than $T_c$ for a low $T_0$, this is precisely what we found here.

The detour via the statistical properties of the overlap between pairs of configurations in supercooled liquids has allowed us to track what remains of the mean-field scenario of glass formation in 2 and 3 dimensions. It would be worth going one step beyond in the direction of building an effective theory for the overlap fluctuations in

finite dimensions by defining a local Franz-Parisi potential over a small region of the sample as the free-energy cost to keep the liquid close a reference configuration in a specific region of space and investigating its fluctuations from one region to another. This would for instance provide access to the local fluctuations of the configurational entropy [104, 105]. This could also help overcoming a limitation of the kind of study presented in this work on the thermodynamics of constrained liquids, which is the lack of a direct connection with the slowdown of relaxation associated with glass formation.

## ACKNOWLEDGMENTS

Some simulations were performed at MESO@LR-Platform at the University of Montpellier. B. Guiselin acknowledges support by Capital Fund Management - Fondation pour la Recherche. This work was supported by a grant from the Simons Foundation (Grant No. 454933, L.B.).

## Appendix A: Analytical results on the $p$-spin model

The fully connected $p$-spin model (with $p \geq 3$) is a paradigmatic example of a mean-field structural glass which has been extensively studied. Our aim is to investigate the influence of the temperature $T_0$ of the reference configurations on the Franz-Parisi potential and on the phase diagram of the constrained system in the $(\epsilon, T)$ plane. For a self-contained presentation we will reproduce derivations and results that are already well-known but which help providing a useful background [60]. The Hamiltonian of the fully connected $p$-spin model is given by

$$\widehat{H}_{\boldsymbol{J}}[\underline{\sigma}] = -\sum_{1 \leq i_1 < \cdots < i_p \leq N} J_{i_1 \ldots i_p} \sigma_{i_1} \ldots \sigma_{i_p}, \qquad \text{(A1)}$$

where $\boldsymbol{J} = \{J_{i_1 \ldots i_p}\}_{1 \leq i_1 < \cdots < i_p \leq N}$ are Gaussian random variables of zero mean and variance $\mathbb{E}\{J_{i_1 \ldots i_p}^2\} = J^2 p!/(2N^{p-1})$, with $J > 0$ a constant that is used as unit energy. In the spherical version which we consider the spin variables are real numbers on the unit sphere, so that spin configurations $\underline{\sigma} = \{\sigma_i\}_{i=1 \ldots N}$ satisfy

$$\frac{1}{N} \sum_{i=1}^{N} \sigma_i^2 = 1. \qquad \text{(A2)}$$

The overlap between a spin configuration $\underline{\sigma}$ and a reference one $\underline{\sigma}^{(0)}$ is

$$\widehat{Q}[\underline{\sigma}; \underline{\sigma}^{(0)}] = \frac{1}{N} \sum_{i=1}^{N} \sigma_i \sigma_i^{(0)}, \qquad \text{(A3)}$$

with, unlike glass-forming liquids, no need to introduce a tolerance lengthscale $a$. The spherical constraint is then merely written as $\widehat{Q}[\underline{\sigma}; \underline{\sigma}] = 1$.

### 1. Cumulants of the (random) Franz-Parisi potential

The Franz-Parisi (FP) potential $V(Q)$, which quantifies the free-energy cost of constraining the overlap $\widehat{Q}[\underline{\sigma}; \underline{\sigma}^{(0)}]$ between two copies $\underline{\sigma}$ and $\underline{\sigma}^{(0)}$ of the same system to a given value $Q$ can be computed exactly, starting from its definition [15],

$$V(Q) = \mathbb{E}\left\{ \int' d\underline{\sigma}^{(0)} \frac{e^{-\beta_0 \widehat{H}_{\boldsymbol{J}}[\underline{\sigma}^{(0)}]}}{\mathcal{Z}_0(\boldsymbol{J})} V(Q; \underline{\sigma}^{(0)}, \boldsymbol{J}) \right\},$$
$$= \mathbb{E}\left\{ \overline{V(Q; \underline{\sigma}^{(0)}, \boldsymbol{J})} \right\}, \qquad \text{(A4)}$$

where $\mathcal{Z}_0(\boldsymbol{J})$ is the partition function at temperature $T_0 = 1/\beta_0$ (the Boltzmann constant is set to unity) for a given realization of the random couplings, the prime symbol on the integral stands for an integration over all the spin configurations which fulfill the spherical constraint, and two distinct averages are introduced: the overline denotes an average over the reference configuration $\underline{\sigma}^{(0)}$ while $\mathbb{E}$ denotes a disorder average over the random couplings $\boldsymbol{J}$. In spite of the additional source of disorder due to the random couplings, the model displays the very same phenomenology for glass formation as mean-field glass-forming liquids [61, 62]. [We will restrict ourselves to reference configurations above the static (Kauzmann) glass transition so that we can assume that the partition function $\mathcal{Z}_0(\boldsymbol{J})$ is self-averaging, hence dropping the dependence on $\boldsymbol{J}$ of the partition function.]

The quantity $V(Q; \underline{\sigma}^{(0)}, \boldsymbol{J})$ is a random function corresponding to the FP potential for a given reference configuration $\underline{\sigma}^{(0)}$ and a given realization $\boldsymbol{J}$ of the random couplings, namely,

$$V(Q; \underline{\sigma}^{(0)}, \boldsymbol{J}) = -\frac{T}{N} \ln \int' d\underline{\sigma} e^{-\beta \widehat{H}_{\boldsymbol{J}}[\underline{\sigma}]} \delta(Q - \widehat{Q}[\underline{\sigma}; \underline{\sigma}^{(0)}]). \qquad \text{(A5)}$$

Its statistical properties can be analyzed through its cumulants. The first cumulant is given by Eq. (A4) and corresponds to the average FP potential. The second one quantifies the total variance of the fluctuations of the FP potential among the realizations of the disorder and is defined as [25, 106, 107]

$$V^{(2)}(Q_1, Q_2) =$$
$$N\beta \left[ \mathbb{E}\left\{ \overline{V(Q_1; \underline{\sigma}^{(0)}, \boldsymbol{J}) V(Q_2; \underline{\sigma}^{(0)}, \boldsymbol{J})} \right\} \right.$$
$$\left. - \mathbb{E}\left\{ \overline{V(Q_1; \underline{\sigma}^{(0)}, \boldsymbol{J})} \right\} \mathbb{E}\left\{ \overline{V(Q_2; \underline{\sigma}^{(0)}, \boldsymbol{J})} \right\} \right], \qquad \text{(A6)}$$

where the factor of $N$ comes from the fact that the FP potential is an intensive quantity and that its typical fluctuations are expected to scale as $N^{-1/2}$, while the factor $\beta$ ensures that $V^{(2)}(Q_1, Q_2)$ has the dimension of an energy. Higher order cumulants $V^{(l)}(Q_1, \ldots, Q_l)$ ($l \geq 3$) can be similarly defined.

In disordered systems, the cumulants can be generated by introducing an arbitrary number $n$ of replicas with the

same realization of the disorder and constrained to have an overlap $\{Q_a\}_{a=1\ldots n}$ with the reference replica 0 and by then considering the replicated FP potential $V_{\rm rep}(\{Q_a\})$ defined through

$$
e^{-N\beta V_{\rm rep}(\{Q_a\})} = \mathbb{E}\left\{\overline{e^{-N\beta\sum_{a=1}^n V(Q_a;\underline{\sigma}^{(0)},\boldsymbol{J})}}\right\}
$$

$$
\propto \mathbb{E}\left\{\int' \prod_{\alpha=0}^n \mathrm{d}\underline{\sigma}^{(\alpha)} e^{-\sum_{\alpha=0}^n \beta_\alpha \widehat{H}_J[\underline{\sigma}^{(\alpha)}]} \times\right.
$$

$$
\left.\prod_{a=1}^n \delta(Q_a - \widehat{Q}[\underline{\sigma}^{(a)};\underline{\sigma}^{(0)}])\right\},
$$
(A7)

with $\beta_a = \beta$ for $1 \le a \le n$. After averaging over the random couplings, this becomes

$$
e^{-N\beta V_{\rm rep}(\{Q_a\})} \propto \int' \prod_{\alpha=0}^n \mathrm{d}\underline{\sigma}^{(\alpha)} e^{\frac{N}{4}\sum_{\alpha,\gamma=0}^n \beta_\alpha \beta_\gamma \widehat{Q}[\underline{\sigma}^{(\alpha)};\underline{\sigma}^{(\gamma)}]^p}
$$

$$
\times \prod_{a=1}^n \delta(Q_a - \widehat{Q}[\underline{\sigma}^{(a)};\underline{\sigma}^{(0)}]).
$$
(A8)

The cumulants can be generated through an expansion in increasing number of sums over replicas [22, 106]:

$$
V_{\rm rep}(\{Q_a\}) = \sum_{a=1}^n V(Q_a) - \frac{1}{2}\sum_{a,b=1}^n V^{(2)}(Q_a,Q_b)
$$

$$
+ \frac{1}{6}\sum_{a,b,d=1}^n V^{(3)}(Q_a,Q_b,Q_d)
$$

$$
- \frac{1}{24}\sum_{a,b,d,e=1}^n V^{(4)}(Q_a,Q_b,Q_d,Q_e) + \cdots
$$
(A9)

The expression in Eq. (A8) can be recast in an integral over all $n\times n$ overlap matrices with diagonal elements equal to 1 (to fulfill the spherical constraint on spin configurations):

$$
e^{-N\beta V_{\rm rep}(\{Q_a\})} \propto \int \prod_{\substack{a,b=1\\a\ne b}}^n \mathrm{d}\widetilde{Q}_{ab} e^{-N\beta\mathcal{V}(\{\widetilde{Q}_{\alpha\gamma}\})}, \quad \text{(A10)}
$$

where we denote $Q_a = \widetilde{Q}_{a0} = \widetilde{Q}_{0a}$ and where the potential $\mathcal{V}(\{\widetilde{Q}_{\alpha\gamma}\})$ is given by

$$
e^{-N\beta\mathcal{V}(\{\widetilde{Q}_{\alpha\gamma}\})} = e^{\frac{N}{4}\sum_{\alpha,\gamma=0}^n \beta_\alpha\beta_\gamma \widetilde{Q}_{\alpha\gamma}^p} \times
$$

$$
\int \prod_{\alpha=0}^n \mathrm{d}\underline{\sigma}^{(\alpha)} \prod_{\alpha,\gamma=0}^n \delta(\widetilde{Q}_{\alpha\gamma} - \widehat{Q}[\underline{\sigma}^{(\alpha)};\underline{\sigma}^{(\gamma)}]).
$$
(A11)

After introducing an exponential representation of the $\delta$-functions and using a saddle-point approximation in the limit of large $N$ [60], one obtains, up to an irrelevant additive constant,

$$
\mathcal{V}(\{\widetilde{Q}_{\alpha\gamma}\}) = -\frac{1}{4}\sum_{\substack{\alpha,\gamma=0\\\alpha\ne\gamma}}^n \beta_\alpha\widetilde{Q}_{\alpha\gamma}^p - \frac{1}{2\beta}\ln\det\widetilde{\boldsymbol{Q}}, \quad \text{(A12)}
$$

with $\widetilde{\boldsymbol{Q}}$ the $(n+1)\times(n+1)$ overlap matrix of elements $\widetilde{Q}_{\alpha\gamma}$. By using another saddle-point approximation for the integration over all overlap matrices, one finds that the replicated FP potential $V_{\rm rep}(\{Q_a\})$ is finally given by an expression of the form of the right-hand side of Eq. (A12) in which the coefficients $\widetilde{Q}_{ab}$ are solution of

$$
\frac{p\beta^2}{4}\widetilde{Q}_{ab}^{p-1} + (\widetilde{\boldsymbol{Q}}^{-1})_{ab} = 0, \quad \text{(A13)}
$$

for $1 \le a,b \le n$ ($a \ne b$). Note that the $\widetilde{Q}_{0a}$'s are fixed (with $\widetilde{Q}_{0a} = \widetilde{Q}_{a0} = Q_a$) and that the solutions of the above equation depend on the $Q_a$'s through the inverse of the matrix $\widetilde{\boldsymbol{Q}}$.

The first cumulant (the average FP potential) can be derived by choosing $Q_a = Q$ for $1 \le a \le n$. By using Eq. (A9) and by only keeping the leading term in the limit $n \to 0$, one finds that $V = \lim_{n\to 0}\partial_n V_{\rm rep}$, where $\partial_n$ denotes the derivative with respect to the number of replicas. To solve Eq. (A13), we insert the 1-step replica symmetry breaking (1-RSB) ansatz with parameters $(\widetilde{Q},Q_0,x)$ for the overlap matrix $\widetilde{Q}_{ab}$, i.e. [60, 108–111],

$$
\widetilde{Q}_{ab} = Q_0 + (\widetilde{Q} - Q_0)\zeta_{ab} + (1 - \widetilde{Q})\delta_{ab}, \quad \text{(A14)}
$$

with $\delta_{ab}$ the identity matrix and $\zeta_{ab}$ the block diagonal matrix with blocks of size $x$ filled with 1. This ansatz is exact at any temperature for $p$-spin models with $p \ge 3$ [58, 112]. The parameters $Q_0$, $\widetilde{Q}$ and $x$ that are involved in the definition of the overlap matrix are solutions of the following saddle-point equations:

$$
\frac{p\beta^2}{2}Q_0^{p-1} = \frac{Q_0 - Q^2}{\left[1 - (1-x)\widetilde{Q} - xQ_0\right]^2}, \quad \text{(A15)}
$$

$$
\frac{p\beta^2}{2}[\widetilde{Q}^{p-1} - Q_0^{p-1}](1-x)
$$

$$
= \frac{(\widetilde{Q} - Q_0)(1-x)}{(1 - \widetilde{Q})\left[1 - (1-x)\widetilde{Q} - xQ_0\right]}, \quad \text{(A16)}
$$

and

$$
\frac{\beta^2}{2}[\widetilde{Q}^p - Q_0^p] + \frac{p\beta^2\widetilde{Q}^{p-1}}{2x}(1 - \widetilde{Q})
$$

$$
- \frac{p\beta^2 Q_0^{p-1}}{2x}\left[1 - (1-x)\widetilde{Q} - xQ_0\right]
$$

$$
+ \frac{1}{x^2}\ln\left[\frac{1 - \widetilde{Q}}{1 - (1-x)\widetilde{Q} - xQ_0}\right] = 0. \quad \text{(A17)}
$$

Furthermore, within the 1-RSB ansatz, the FP potential

reads

$$V_{\text{RSB}}(Q) = -\frac{\beta_0}{2}Q^p + \frac{\beta}{4}\left[(1-x)\widetilde{Q}^p + xQ_0{}^p\right]$$
$$+ \frac{1}{2\beta}\frac{1-x}{x}\ln(1-\widetilde{Q})$$
$$- \frac{1}{2\beta x}\ln\left[1 - (1-x)\widetilde{Q} - xQ_0\right] \qquad \text{(A18)}$$
$$- \frac{Q_0 - Q^2}{2\beta\left[1 - (1-x)\widetilde{Q} - xQ_0\right]}.$$

The simpler replica-symmetric (RS) case, which gives the correct solution of Eq. (A13) at high-enough temperatures [15, 58, 113] is easily obtained from the 1-RSB expression by setting $\widetilde{Q} = Q_0$, leading to

$$V_{\text{RS}}(Q) = -\frac{\beta_0}{2}Q^p + \frac{\beta}{4}\widetilde{Q}^p - \frac{1}{2\beta}\ln(1-\widetilde{Q}) - \frac{\widetilde{Q}-Q^2}{2\beta(1-\widetilde{Q})},$$
$$\text{(A19)}$$

where $\widetilde{Q} \equiv \widetilde{Q}(Q)$ satisfies

$$\frac{p\beta^2}{2}\widetilde{Q}^{p-1} = \frac{\widetilde{Q}-Q^2}{(1-\widetilde{Q})^2}. \qquad \text{(A20)}$$

At this point, we note that the saddle-point equations [Eqs. (A15)-(A17) or Eq. (A20)] do not depend on $T_0$, and their solution can thus be computed at once for the case $T = T_0$. The FP potential itself nonetheless depends on $T_0$ through the first term in the right-hand side, and

$$V(Q) = V^{(T=T_0)}(Q) + \frac{\beta - \beta_0}{2}Q^p, \qquad \text{(A21)}$$

so that the FP potential for any temperature of the reference configurations can be straightforwardly obtained from its value when $T = T_0$.

The second cumulant can be computed by introducing two groups of replicas: $n_1$ replicas having an overlap $Q_1$ with the reference configuration and $n_2$ having an overlap $Q_2$ with the reference configuration (with $n_1 + n_2 = n$). Using Eq. (A9), one has that $V^{(2)} = -\lim_{n_1,n_2\to 0}\partial_{n_1}\partial_{n_2}V_{\text{rep}}$. In the following, we only consider the vicinity of the critical point in the $(\epsilon, T)$ plane and we will verify that it is always in the replica-symmetric region. This leads to

$$V_{\text{RS}}^{(2)}(Q_1, Q_2) = \frac{\beta}{2}Q_{12}{}^p - \frac{(Q_{12} - Q_1 Q_2)^2}{2\beta(1-\widetilde{Q}_1)(1-\widetilde{Q}_2)}, \quad \text{(A22)}$$

where $\widetilde{Q}_a$ $(a = 1, 2)$ are solutions of Eq. (A20) with $Q$ replaced by $Q_a$ and $Q_{12} \equiv Q_{12}(Q_1, Q_2)$ is an extremum of Eq. (A22), i.e.,

$$\frac{p\beta^2}{2}Q_{12}{}^{p-1} = \frac{Q_{12} - Q_1 Q_2}{(1-\widetilde{Q}_1)(1-\widetilde{Q}_2)}. \qquad \text{(A23)}$$

We note that neither the solution of Eq. (A23) nor the expression in Eq. (A22) depend on the temperature $T_0$ of the reference configurations. In addition, one finds that if $Q_1 = Q_2$ (in particular at the critical point), then $Q_{12} = \widetilde{Q}_1 = \widetilde{Q}_2$.

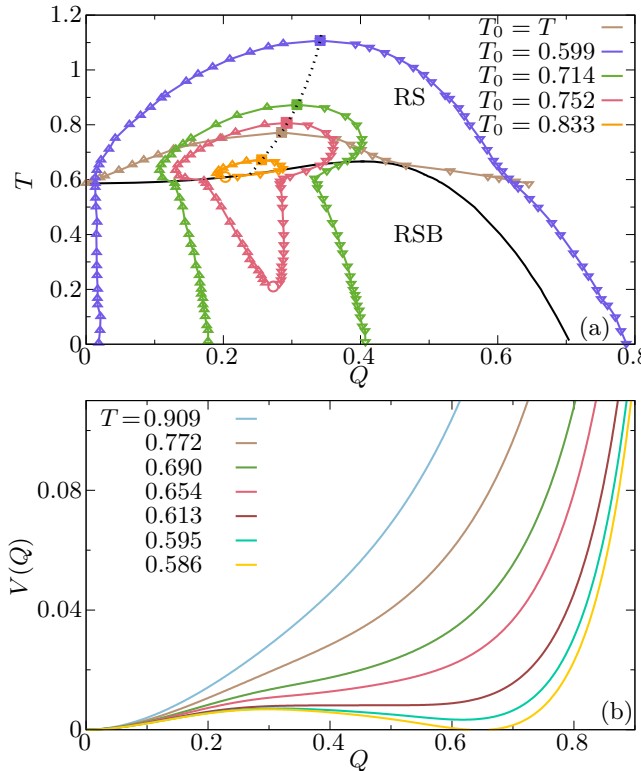

FIG. 12. Franz-Parisi (FP) construction for the fully connected spherical $p$-spin model with $p = 3$. (a) One-step replica symmetry breaking (RSB) and replica-symmetric (RS) regions obtained by solving the saddle-point equations [Eqs. (A15)-(A17)] for each pair $(Q, T)$. The full black line delimiting the two regions represents the discontinuous and continuous RSB transitions. We have also reported the values of $Q$ in the low-overlap (up triangles) and high-overlap (down triangles) phases at coexistence in the first-order transition region, corresponding to the two minima of equal height of the tilted FP potential $V_\epsilon(Q) = V(Q) - \epsilon Q$ when $T_0 = T$ and for several values of $T_0$ fixed. The square marks the location of the critical point $(Q_c, T_c)$, where $Q_c$ stands for the critical overlap. The dotted line represents the loci of $Q_c$ as a function of $T_0$: see the calculations of Sec. A 4. (b) FP potential for $T = T_0$ and $p = 3$. The FP potential is strictly convex at high temperatures and loses convexity at $T_{\text{cvx}} = 0.772$ ($\beta_{\text{cvx}} = 1.295$). A metastable minimum appears at the dynamical transition temperature $T_d = 0.613$ ($\beta_d = 1.632$) and the two minima become equally stable at the static (Kauzmann) transition temperature $T_K = 0.586$ ($\beta_K = 1.707$).

## 2. Evolution with temperature of the Franz-Parisi potential

The Franz-Parisi (FP) potential can be numerically computed for any temperature by solving Eqs. (A15)-(A17) for increasing values of $Q \in [0, 1]$ and by finally using Eq. (A18). When $T \leq T_{\text{RSB}}$ (= 0.666 for $p = 3$), the replica symmetry is broken for intermediate values of the overlap $Q \in [Q_{\text{min,RBS}}(T), Q_{\text{max,RSB}}(T)]$ whose range increases as the temperature decreases: see Fig. 12(a). For $T \leq T_K$, the replica symmetry becomes broken even

in the minimum at $Q = 0$. A discontinuous replica symmetry breaking occurs at $Q = Q_{\mathrm{min,RBS}}(T)$ (with a jump in $\widetilde{Q}$ as a function of $Q$) and a continuous one at $Q = Q_{\mathrm{max,RSB}}(T)$.

The evolution with the temperature of the FP potential for the case $T = T_0$ is illustrated in Fig. 12(b). This result is already well known [17]. At high-enough temperatures, the FP potential is convex with a single minimum for $Q = 0$ down to the temperature $T_{\mathrm{cvx}}$ at which it first loses its convexity. A second minimum appears at a lower temperature $T_d$, which also corresponds to the dynamical glass transition in which the system gets trapped in a metastable glassy state. Below $T_d$ the difference in height between the secondary minimum and the stable one is the free-energy cost to maintain the replicas in the same metastable state and therefore provides the configurational entropy per spin $s_c(T)$ related to the logarithm of the number of metastable states (which are well-defined in this mean-field limit). At a still lower temperature $T_K$, the configurational entropy vanishes and a random first-order phase transition takes place between the liquid at $Q = 0$ and the ideal glass at $Q = Q_g > 0$.

### 3. Phase diagrams in the $(\epsilon, T)$ plane

Whenever the Franz-Parisi (FP) potential is not convex, a well-chosen nonzero source $\epsilon$ linearly coupled to the overlap $Q$ can tilt the FP potential so that $V_\epsilon(Q) = V(Q) - \epsilon Q$ has a double-well structure with two minima of equal depth, inducing a first-order phase transition between a low-overlap phase at high temperature and small $\epsilon$ (delocalized phase) and a high-overlap phase at low temperature and large $\epsilon$ (localized phase) [16, 17, 114–116]. The phase diagram for the case $T = T_0$ obtained from the double tangent construction is shown in Fig. 2(a) of the main text and is also reproduced in Fig. 13(a). A line of first-order transition emerges from the random first-order transition (RFOT) point at $(0, T_K)$ and ends in a critical point $(\epsilon_c^{(T=T_0)}, T_c^{(T=T_0)})$ at the temperature $T_c^{(T=T_0)} = T_{\mathrm{cvx}}$ at which the FP potential first loses convexity [16]. We also report in Fig. 12(a) the values of the overlap in the low- and high-overlap phases obtained from the double tangent construction for $T \leq T_c^{(T=T_0)}$, and we note that both always lie in the replica-symmetric region, except at $T_K$.

We study the influence of the temperature $T_0$ of the reference configurations (with $T_0 \geq T_K$). It is known that the FP potential has a secondary minimum in the temperature range $0 < T < T_f(T_0)$ as long as $T_0 < T_d$ [25, 113]. When this minimum exists, its height has two contributions, one coming from the entropic cost for selecting a particular metastable state at the temperature $T_0$, the other from the difference between the free energy of the metastable states that dominate at $T_0$ and are followed to the temperature $T$ and the equilibrium free energy at the temperature $T$ [15, 113]. It has also been

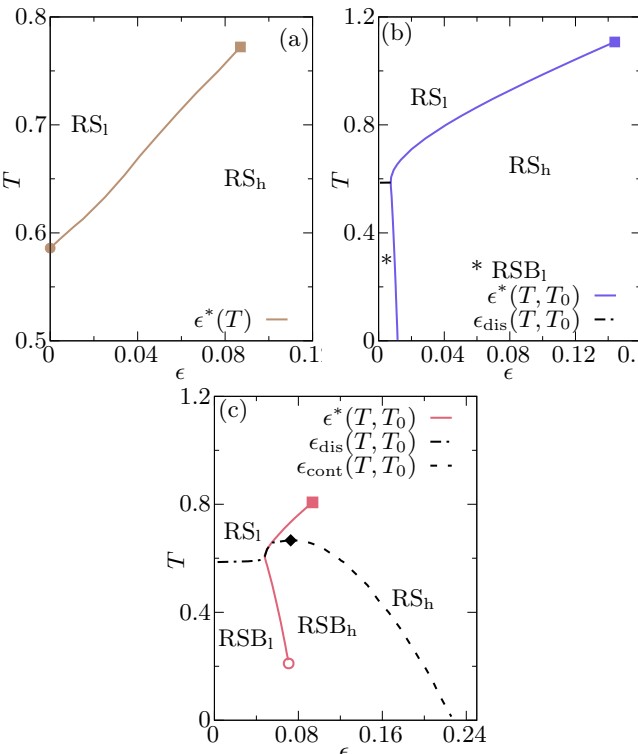

FIG. 13. Phase diagram of the fully connected spherical $p$-spin model with $p = 3$ in the $(\epsilon, T)$ plane for different cases concerning the temperature $T_0$ of the reference configurations: (a) $T = T_0$, (b) $T_0 < T_d$ ($T_0 = 0.599$ or $\beta_0 = 1.67$) and (c) $T_d < T_0 < T_{\mathrm{cvx}}$ ($T_0 = 0.752$ or $\beta_0 = 1.33$). We have displayed the line of first-order transition $\epsilon^*(T, T_0)$ from the delocalized phase to the localized phase, along with the lines of continous, $\epsilon_{\mathrm{con}}(T, T_0)$, and discontinuous, $\epsilon_{\mathrm{dis}}(T, T_0)$, replica symmetry breaking (RSB) transitions. The phase diagrams display at most four different phases: a low-overlap replica-symmetric (RS) phase (RS$_l$), a low-overlap one-step RSB (1-RSB) phase (RSB$_l$), a high-overlap RS phase (RS$_h$), and a high-overlap 1-RSB phase (RSB$_h$). In all panels, the full square marks the position of the high-temperature critical point, the empty disk the end of the first-order transition line at a low temperature, the full disk the static glass transition at $(\epsilon = 0, T_K)$, and the full diamond the temperature at which RSB effects appear ($T_{\mathrm{RSB}} = 0.666$ or $\beta_{\mathrm{RSB}} = 1.502$). The overlap is discontinuous on the line $\epsilon^*(T)$ or $\epsilon^*(T, T_0)$ but is continuous otherwise.

found that when $T_0 > T_d$, the FP potential no longer displays a secondary minimum, whatever the temperature $T$.

We display in Fig. 2(a) of the main text the phase diagram for a fixed $T_0$ between $T_d$ and $T_K$ ($T_0 = 0.599$ or $\beta_0 = 1.67$). It is reproduced in Fig. 13(b), where we additionally show the replica symmetry breaking (RSB) transitions. The phase diagram shows some differences with the case $T = T_0$. The main one is that the first-order transition line does not converge to the RFOT point at $T_K$ but instead strongly bends and goes to zero temperature for a finite value of $\epsilon$. The first-order transition line still ends in a critical point which appears to be shifted

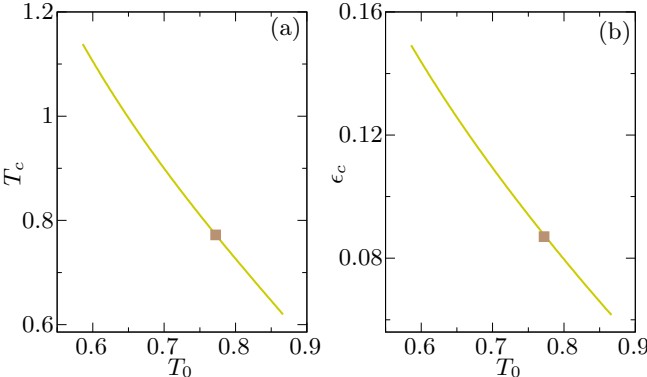

FIG. 14. Evolution of the location of the critical point as a function of the temperature $T_0$ of the reference configurations for the fully connected spherical $p$-spin model with $p = 3$: (a) critical temperature $T_c$ and (b) critical source $\epsilon_c$. The critical point only exists when it is in the replica-symmetric phase. The square marks the location of the critical point when $T = T_0$. The Kauzmann temperature is at $T_K = 0.586$.

up in temperature and in applied source (see below). We note that the continuous RSB transition is absent, as the high-overlap phase is always replica-symmetric, as seen in Fig. 12(a). There is however a discontinuous RSB transition close to $T_K$. We recall that the overlap is continuous at this transition while the saddle-point solution $\widetilde{Q}$ is not.

For the sake of completeness, we have also studied the intermediate case where $T_d < T_0 < T_{\text{cvx}}$. This is illustrated in Fig. 13(c). The critical point still seems to be shifted upward in $T$ and $\epsilon$. If $T_0 < T_z$ ($\approx 0.749$ for $p = 3$), the line of first-order transition still ends at zero temperature and finite source. However, if $T_0 > T_z$, the line ends in another critical point at a low temperature in the 1-RSB region. We find four different phases in the diagram, as illustrated for $T_0 > T_z$ in Fig. 13(c). We have finally investigated the case where $T_0 > T_{\text{cvx}}$. It also leads to a complex pattern of RSB transitions but this it is not directly relevant to the physical situation that we are interested in and we do not show the results. [This is indirectly displayed in Fig. 12(a): see the curve for $T_0 = 0.833$.] In particular, we found that the critical point at high temperature disappears when it enters the 1-RSB region.

### 4. Variation of the location of the critical point with the temperature $T_0$ of the reference configurations

To systematically study the location of the critical point $(\epsilon_c, T_c)$ when varying $T_0$, we use the replica-symmetric (RS) expression of the Franz-Parisi (FP) potential given by Eq. (A19). Indeed, we have already mentioned that the critical point disappears when it enters the region of replica symmetry breaking. To simplify no-

tations, we now drop the subscript RS. To find the critical point, we need to solve the set of equations

$$
\begin{aligned}
V'(Q_c) &= \epsilon_c, \\
V''(Q_c) &= 0, \\
V'''(Q_c) &= 0,
\end{aligned}
\tag{A24}
$$

for the triplet $(Q_c, \epsilon_c, T_c)$, with $Q_c$ the value of the overlap at the critical point. Physically, the last two equations are equivalent to requiring that the isotherm ($\epsilon$ as a function of the average overlap) has an inflexion point with a horizontal tangent line.

The derivatives in Eq. (A24) can be computed from Eq. (A19) and Eq. (A20), the latter being used to obtain the derivatives of the saddle-point solution $\widetilde{Q}(Q)$ with respect to $Q$. The first derivative of the FP potential reads

$$
V'(Q) = -\frac{p\beta_0}{2}Q^{p-1} + \frac{Q}{\beta[1 - \widetilde{Q}(Q)]},
\tag{A25}
$$

where we have used that the derivative of Eq. (A19) with respect to $\widetilde{Q}$ is zero due to the saddle-point condition. The second derivative can be found in the same way:

$$
\begin{aligned}
V''(Q) = &-\frac{p(p-1)\beta_0}{2}Q^{p-2} + \frac{1}{\beta[1 - \widetilde{Q}(Q)]} \\
&+ \frac{Q}{\beta[1 - \widetilde{Q}(Q)]^2}\widetilde{Q}'(Q),
\end{aligned}
\tag{A26}
$$

where the first derivative $\widetilde{Q}'(Q)$ of the saddle-point solution with respect to $Q$ can be obtained by differentiating the saddle-point equation (A19) with respect to $Q$:

$$
\widetilde{Q}'(Q) = \frac{-2Q}{(p\beta^2/2)[\widetilde{Q}(Q)]^{p-2}[1 - \widetilde{Q}(Q)][p - 1 - (p+1)\widetilde{Q}(Q)] - 1}.
\tag{A27}
$$

The third derivative is obtained by using the same procedure. It involves the second derivative of the saddle-point equation with respect to $Q$, which can be expressed by differentiating Eq. (A27) with respect to $Q$. The resulting expressions are not reproduced here.

We display in Fig. 14 the evolution of $T_c$ and $\epsilon_c$ with the temperature $T_0$ of the reference configurations. When $T_0$ is fixed, the critical temperature $T_c$ is a monotonically decreasing function of $T_0$. The figure clearly shows that when $T_0$ is fixed to a temperature below $T_c^{(T=T_0)} = T_{\text{cvx}}$, the critical point is shifted upward in temperature and in $\epsilon$ in the phase diagram. By contrast, when $T_0$ is fixed above $T_{\text{cvx}}$, the critical temperature and critical source are shifted downward (until replica symmetry becomes broken). This feature can be easily understood from Eq. (A21). We note that the second term in the right-hand side is positive if $T < T_0$ and negative otherwise. As $T_{\text{cvx}}$ is the highest temperature at which $V^{(T=T_0)}(Q)$ develops an inflexion point, taking $T_0$ smaller (respectively,

larger) than $T_{\rm cvx}$ makes the FP potential at $T = T_{\rm cvx}$ even more nonconvex (respectively, convex), pushing the critical critical point up (respectively, down) in temperature. The same observations hold for $\epsilon_c(T_0)$, suggesting that when $T_0$ decreases the critical source has to overcome larger thermal fluctuations in order for the system to fall in the localized phase, as $T_c$ also increases. Note that the case where $T_0 > T_{\rm cvx}$ is not relevant to our study and is not easily interpretable in terms of the physics of glass-forming liquids.

The variation of the critical temperature is quite large, of about 25 % between the case $T = T_0$ and the case of fixed $T_0 = T_K$. As a result, by considering the overlap with an equilibrium reference configuration sampled at a very low temperature (but still above the Kauzmann transition), it is possible to move the critical point high up in the liquid region. We expect this feature to persist in finite dimensions, thus motivating our choice of very stable reference configurations prepared with the help of the swap Monte Carlo algorithm for the numerical study described in the main text.

### 5. Beyond mean-field: effective Landau-Ginzburg action in the vicinity of the critical point

Following the analysis of Ref. [22], we introduce finite-dimensional fluctuations of the overlap in the spherical $p$-spin model by building an effective Landau-Ginzburg action in the vicinity of the (mean-field) critical point, but contrary to Ref. [22] that was focused on the case $T = T_0$, we consider the generic situation of a fixed temperature $T_0$ of the reference configurations.

The local part of the action is obtained by performing a Taylor expansion of the replicated Franz-Parisi (FP) potential $\beta V_{\rm rep}(\{Q_a\}; \beta, \beta_0)$, where we have explicitly displayed the dependence on $\beta = 1/T$ and $\beta_0 = 1/T_0$, for $Q_a = Q_c + \phi_a$. Up to an irrelevant additive constant, this gives in the vicinity of the mean-field critical point $(\beta_c, \epsilon_c)$

$$
\beta V_{\rm rep}(\{Q_a\}; \beta, \beta_0) - \beta\epsilon \sum_{a=1}^{n} Q_a = \sum_{a=1}^{n} \Big[ \frac{g_2}{2}{\phi_a}^2 + \frac{g_3}{6}{\phi_a}^3 +
$$

$$
\frac{g_4}{24}{\phi_a}^4 \Big] - \frac{1}{2} \sum_{a,b=1}^{n} \phi_a \phi_b \Big[ \tau_{20} + \frac{\tau_{21}}{2} (\phi_a + \phi_b) + \frac{\tau_{22}}{4}\phi_a\phi_b
$$

$$
+ \frac{\tau_{23}}{6} \left({\phi_a}^2 + {\phi_b}^2\right) \Big] + \frac{1}{6} \sum_{a,b,d=1}^{n} \phi_a\phi_b\phi_d \Big[ \tau_{30} + \frac{\tau_{31}}{2}\big(\phi_a
$$

$$
+ \phi_b + \phi_d\big) \Big] - \frac{\tau_{40}}{24} \sum_{a,b,d,e=1}^{n} \phi_a\phi_b\phi_d\phi_e + \dots ,
$$

$$(A28)$$

where the coefficients involved in the expansion can be expressed in terms of derivatives of the cumulants of the FP potential [22]; for instance,

$$
g_2 = \beta_c V''(Q_c; \beta_c, \beta_0), \ g_3 = \beta_c V'''(Q_c; \beta, \beta_0), \quad (A29)
$$

which both vanish at the (mean-field) critical point, and, from higher cumulants,

$$
\begin{aligned}
\tau_{20} &= \beta_c \partial_{Q_1}\partial_{Q_2} V^{(2)}(Q_1, Q_2; \beta_c, \beta_0)|_{Q_c}, \\
\tau_{21} &= \beta_c \partial_{Q_1}^2 \partial_{Q_2} V^{(2)}(Q_1, Q_2; \beta_c, \beta_0)|_{Q_c}, \\
\tau_{22} &= \beta_c \partial_{Q_1}^2 \partial_{Q_2}^2 V^{(2)}(Q_1, Q_2; \beta_c, \beta_0)|_{Q_c}, \\
\tau_{23} &= \beta_c \partial_{Q_1}^3 \partial_{Q_2} V^{(2)}(Q_1, Q_2; \beta_c, \beta_0)|_{Q_c}, \\
\tau_{30} &= \beta_c \partial_{Q_1}\partial_{Q_2}\partial_{Q_3} V^{(3)}(Q_1, Q_2, Q_3; \beta_c, \beta_0)|_{Q_c},
\end{aligned}
$$

$$(A30)$$

etc.

The effective Landau-Ginzburg action should allow for nonuniform overlap profiles and include a penalty for too strong fluctuations between low- and high-overlap regions. This can be done by considering a Kac version of the spherical $p$-spin model [117, 118], as in Ref. [47], but a short-cut is to envisage an expansion in spatial gradients of the overlap field and to keep only the lowest-order term. The resulting effective action reads

$$
\mathcal{S}_{\rm rep,eff}(\{\phi_a\}; \beta, \beta_0) = \sum_{a=1}^{n} \int {\rm d}^d\boldsymbol{x}\big[ K\left(\partial_{\boldsymbol{x}}\phi_a(\boldsymbol{x})\right)^2 +
$$

$$
\frac{g_2}{2}\phi_a(\boldsymbol{x})^2 + \frac{g_3}{6}\phi_a(\boldsymbol{x})^3 + \frac{g_4}{24}\phi_a(\boldsymbol{x})^4\big] - \frac{1}{2}\sum_{a,b=1}^{n} \int {\rm d}^d\boldsymbol{x}
$$

$$
\phi_a(\boldsymbol{x})\phi_b(\boldsymbol{x})\big[\tau_{20} + \frac{\tau_{21}}{2}\left(\phi_a(\boldsymbol{x}) + \phi_b(\boldsymbol{x})\right) + \frac{\tau_{22}}{4}\phi_a(\boldsymbol{x})\phi_b(\boldsymbol{x})
$$

$$
+ \frac{\tau_{23}}{6}\left(\phi_a(\boldsymbol{x})^2 + \phi_b(\boldsymbol{x})^2\right)\big] + \frac{1}{6}\sum_{a,b,d=1}^{n} \int {\rm d}^d\boldsymbol{x}\phi_a(\boldsymbol{x})\times
$$

$$
\phi_b(\boldsymbol{x})\phi_d(\boldsymbol{x})\big[\tau_{30} + \frac{\tau_{31}}{2}\left(\phi_a(\boldsymbol{x}) + \phi_b(\boldsymbol{x}) + \phi_d(\boldsymbol{x})\right)\big]
$$

$$
- \frac{\tau_{40}}{24}\sum_{a,b,d,e=1}^{n} \int {\rm d}^d\boldsymbol{x}\phi_a(\boldsymbol{x})\phi_b(\boldsymbol{x})\phi_d(\boldsymbol{x})\phi_e(\boldsymbol{x}) + \cdots ,
$$

$$(A31)$$

where $K > 0$ is a phenomenological parameter, $\partial_{\boldsymbol{x}}$ denotes a spatial gradient, and the ellipses denote higher-order terms in the number of replicas, fields and/or gradients.

It is then possible to show that the above effective Landau-Ginzburg functional can be mapped onto the replicated Hamiltonian of a system in the presence of a random field $h(\boldsymbol{x})$, a random mass $m(\boldsymbol{x})$ and a random cubic coupling $\lambda(\boldsymbol{x})$ [22], whose disordered Hamiltonian is

$$
\beta\mathcal{H}[\phi(\boldsymbol{x})] = \int {\rm d}^d\boldsymbol{x}\big[ K\left(\partial_{\boldsymbol{x}}\phi(\boldsymbol{x})\right)^2 + \frac{g_2}{2}\phi(\boldsymbol{x})^2 + \frac{g_3}{6}\phi(\boldsymbol{x})^3
$$

$$
+ \frac{g_4}{24}\phi(\boldsymbol{x})^4 + \frac{m(\boldsymbol{x})}{2}\phi(\boldsymbol{x})^2 + \frac{\lambda(\boldsymbol{x})}{6}\phi(\boldsymbol{x})^3 - h(\boldsymbol{x})\phi(\boldsymbol{x})\big],
$$

$$(A32)$$

where the random field, random mass, and random cou-

pling have zero mean and higher cumulants given by

$$
\begin{aligned}
\overline{h(\boldsymbol{x})h(\boldsymbol{y})} &= \tau_{20}\delta^{(d)}(\boldsymbol{x}-\boldsymbol{y}), \\
\overline{h(\boldsymbol{x})m(\boldsymbol{y})} &= -\tau_{21}\delta^{(d)}(\boldsymbol{x}-\boldsymbol{y}), \\
\overline{m(\boldsymbol{x})m(\boldsymbol{y})} &= \tau_{22}\delta^{(d)}(\boldsymbol{x}-\boldsymbol{y}), \\
\overline{h(\boldsymbol{x})\lambda(\boldsymbol{y})} &= -\tau_{23}\delta(\boldsymbol{x}-\boldsymbol{y}), \\
\overline{h(\boldsymbol{x})h(\boldsymbol{y})h(\boldsymbol{t})} &= -\tau_{30}\delta^{(d)}(\boldsymbol{x}-\boldsymbol{y})\delta^{(d)}(\boldsymbol{x}-\boldsymbol{t}),
\end{aligned}
\tag{A33}
$$

etc., where an overline denotes the disorder average while $\delta^{(d)}$ stands for the Dirac distribution in $d$ dimensions. Consistency of the mapping requires that $\tau_{20} > 0$, $\tau_{22} > 0$, etc.

In the absence of spin-glass-like frustrating interactions, provided $\tau_{20} > 0$, the above disordered system is known to be in the universality class of the RFIM. A short-range correlated random field $h(x)$ that breaks the $Z_2$ inversion symmetry in any given sample (the symmetry is only statistically recovered after disorder-averaging) and the 1-replica $\phi^4$-theory are the necessary ingredients for this universality class: the other disorder terms as well as additional gradient terms describing nonlocal but short-ranged behavior or terms associated with higher-order cumulants of the disorder are indeed generated along the renormalization-group flow, even in the standard RFIM [106]. This for the exact same reason that the whole Ising critical universality class can be described by starting from the Wilson-Ginzburg-Landau $\phi^4$-theory.

The above derivation therefore shows that, if the critical point survives in finite dimensions, it is in the universality class of the RFIM. Its lower critical dimension is then $d_l = 2$ and in $3d$ it may survive if the strength of the disorder is not too strong [46]. These considerations are expected to apply even when considering finite-dimensional realistic supercooled liquids.

For the spherical $p$-spin, the parameters of the effective random system can be explicitly obtained for any temperature $T_0$ of the reference configurations. We focus on $\tau_{20}$, which represents the effective strength of the random field and can be computed from the second cumulant of the FP potential given in Eq. (A22). This yields

$$
\begin{aligned}
\tau_{20}(T_0) =\ & \frac{-Q_c^2}{[1-\widetilde{Q}(Q_c)]^2} + \frac{p\beta_c^2}{2}\widetilde{Q}(Q_c)^{p-1} \\
& + \frac{Q_c\widetilde{Q}'(Q_c)}{2[1-\widetilde{Q}(Q_c)]^2}\left\{1 + p\beta_c^2\widetilde{Q}(Q_c)^{p-1}[1-\widetilde{Q}(Q_c)]\right\} \\
& - \frac{p\beta_c^2\widetilde{Q}(Q_c)^{p-1}\widetilde{Q}'(Q_c)^2}{4[1-\widetilde{Q}(Q_c)]} \\
& \times \left\{1 + \frac{p\beta_c^2}{2}\widetilde{Q}(Q_c)^{p-1}[1-\widetilde{Q}(Q_c)]\right\},
\end{aligned}
\tag{A34}
$$

where the dependence on $T_0$ comes from that of $Q_c$ and $T_c$. Besides the derivatives of $Q_{12}(Q_1, Q_2)$ with respect to $Q_1$ or $Q_2$ at the critical point have been expressed as a function of $\tilde{Q}'(Q_c)$.

The evolution of the random-field variance $\Delta \equiv \tau_{20}$ with the temperature $T_0$ of the reference configurations is shown in Fig. 2(b) of the main text: $\Delta$ decreases at both large and small values of $T_0$ while it is maximum for intermediate values with $T_0 \approx T_{\mathrm{cvx}}$. This in particular implies that the case $T_0 = T$ corresponds to a relatively high random-field disorder strength.

## Appendix B: Models and methods

### 1. Models

We study a system of $N$ spherical particles of equal mass $m$ in spatial dimensions $d = 2$ and $d = 3$ with radial pairwise interactions, as first introduced in Ref. [55]. The diameters $\{\sigma_i\}_{i=1\dots N}$ of the particles are drawn from the distribution $p(\sigma_i) \propto \sigma_i^{-3}$ for $\sigma_i \in [\sigma_{\min}, \sigma_{\max}]$ with $\sigma_{\max}/\sigma_{\min} \approx 2.217$. Two particles $i$ and $j$ interact with the repulsive potential

$$
v(r_{ij}) = v_0\left(\frac{\sigma_{ij}}{r_{ij}}\right)^{12} + c_0 + c_2\left(\frac{r_{ij}}{\sigma_{ij}}\right)^2 + c_4\left(\frac{r_{ij}}{\sigma_{ij}}\right)^4
\tag{B1}
$$

if their relative distance $r_{ij} = |\boldsymbol{r}_i - \boldsymbol{r}_j|$ satisfies $r_{ij}/\sigma_{ij} < x_c = 1.25$; $v_0$ is the interaction strength and the interaction cross-diameter $\sigma_{ij}$ is given by the nonadditive rule ($\mu > 0$)

$$
\sigma_{ij} = \frac{\sigma_i + \sigma_j}{2}(1 - \mu|\sigma_i - \sigma_j|).
\tag{B2}
$$

The constants $c_0$, $c_2$ and $c_4$ are set in order to make the potential and its two first derivatives continuous at the cut-off distance $x_c$: $c_0 = -28v_0/x_c^{12}, c_2 = 48v_0/x_c^{14}, c_4 = -21v_0/x_c^{16}$. The distribution of diameters along with the nonadditive rule for cross-diameters reduce the tendency of the system for crystallization or demixing. The average diameter $\sigma$ of the particles is used as unit length ($\mu = 0.2$ in this unit), the interaction strength $v_0$ is used as unit temperature (the Boltzmann constant $k_B$ is set to unity), and $\sqrt{m\sigma^2/v_0}$ is used as unit time. The system is simulated in a cubic box of linear size $L$ with periodic boundary conditions [119]. The number density $\rho = N/L^d$ is chosen equal to 1.

The unconstrained liquid is simulated by using a hybrid scheme combining molecular dynamics in the canonical ensemble (NVT-MD) and the recently developed swap Monte Carlo algorithm in order to speed up equilibration and exploration of the phase space [56]. The scheme consists in the succession of blocks of MD steps separated by blocks during which swap moves are performed. The MD is run by implementing the Hoover equations [120] of the Nosé thermostat [121–123] with a time step $dt$ and a thermostat damping time $\tau_{\mathrm{th}}$ (see Table I). The equations of motion are integrated by means of a reversible integrator based on a Liouville formulation of the equations [124, 125]. The MD is run for $n_{\mathrm{MD}}$ steps (see Table I). Then, the positions and velocities

| | d$t$ | $\tau_{\text{th}}$ | $n_{\text{MD}}$ | $n_{\text{swap}}$ | $a$ | $\kappa$ | $T_0$ | $T_{\text{mct}}$ | $T_g$ |
|---|---|---|---|---|---|---|---|---|---|
| $2d$ | 0.005 | 0.5 | 50 | 10 | 0.22 | 0.3 | 0.03 | 0.115 | 0.068 |
| $3d$ | 0.01 | 0.5 | 25 | 1 | 0.22 | 20 | 0.06 | 0.095 | 0.056 |

TABLE I. Parameters used to run the simulations: time step d$t$ for the integration of the equations of motion, damping time of the thermostat $\tau_{\text{th}}$, number of molecular dynamics steps $n_{\text{MD}}$ between sequences of swap moves, number of swap moves per particle $n_{\text{swap}}$, tolerance length $a$ in the definition of the overlap, curvature $\kappa$ of the umbrella potential, fixed temperature $T_0$ of the reference configurations. We also report the mode-coupling crossover temperature $T_{\text{mct}}$ and the extrapolated calorimetric glass transition temperature $T_g$ for comparison [63, 65].

of the particles are frozen and $N_{\text{swap}} = n_{\text{swap}}N$ swap moves are attempted (see Table I). For an elementary swap move, two particles $i$ and $j$ are randomly selected and their diameters are exchanged. The change in the total potential energy $\Delta\widehat{H}_{\text{swap}} = \widehat{H}_{\text{swap}} - \widehat{H}$ is then computed with $\widehat{H}$ given by Eq. (2) (as the kinetic energy remains constant) and $\widehat{H}_{\text{swap}}$ the total potential energy when particle diameters are swapped. The move is eventually accepted following the Metropolis rule, *i.e.*, with probability $\min(1, e^{-\beta\Delta\widehat{H}_{\text{swap}}})$ (with $\beta = 1/T$), in order to guarantee detailed balance [73, 119]. This combination of NVT-MD and swap moves ensures a proper sampling in the canonical ensemble. The parameters $n_{\text{MD}}$ and $n_{\text{swap}}$ have been chosen to maximize the algorithm efficiency.

To compute the overlap between two configurations [see Eq. (1)], we use the window function $w(x) = e^{-x^4 \ln 2}$ with a tolerance length $a$ reported in Table I. The influence of the tolerance length on the results presented in the main text was extensively studied in Ref. [33] where we focused on a mean-field-like model, the hypernetted chain approximation of liquid-state theory (see Refs. [126–128] and Refs. [18, 19, 30–32] for its application in the Franz-Parisi setting). It was found that the qualitative features of the phase diagram in the $(\epsilon, T)$ plane are insensitive to the choice of $a$, even though the precise location of the critical point is quantitatively changed when varying $a$. Here we have chosen a relatively small value of $a$, *i.e.*, $a = 0.22$.

### 2. Umbrella sampling

From Eq. (3), it is obvious that when a source $\epsilon$ is applied, the probability distribution of the overlap for a fixed reference configuration $\boldsymbol{r}_0^N$ is simply given by $\mathcal{P}_\epsilon(Q; \boldsymbol{r}_0^N) \propto \mathcal{P}(Q; \boldsymbol{r}_0^N)e^{N\beta\epsilon Q}$ where $\mathcal{P}(Q; \boldsymbol{r}_0^N) = \mathcal{P}_{\epsilon=0}(Q; \boldsymbol{r}_0^N)$ is the probability distribution of the overlap in the unconstrained liquid at a temperature $T$. As a result, to accurately compute thermodynamic quantities for any $\epsilon$, overlap fluctuations in the unconstrained liquid with an exponentially small weight in $N$ must be

measured. In a conventional simulation, the system typically explores a narrow range of overlap values around the random value $Q_{\text{rand}}$, which corresponds to the overlap for two uncorrelated configurations and which is the absolute minimum of the Franz-Parisi potential. Consequently, a good measure of $\mathcal{P}(Q; \boldsymbol{r}_0^N)$ on the entire range of overlap requires a sophisticated algorithm to sample rare events.

We use umbrella sampling [69–71] to force the unconstrained liquid toward large and untypical values of the overlap and we sample the phase space with the biased Hamiltonian

$$\widehat{H}_{\text{b}}[\boldsymbol{r}^N; \boldsymbol{r}_0^N] = \widehat{H}[\boldsymbol{r}^N] + W(\widehat{Q}[\boldsymbol{r}^N; \boldsymbol{r}_0^N])$$
$$= \widehat{H}[\boldsymbol{r}^N] + \frac{1}{2}N\kappa(\widehat{Q}[\boldsymbol{r}^N; \boldsymbol{r}_0^N] - Q^{\text{ref}})^2 \quad \text{(B3)}$$

which is obtained by adding a harmonic bias $W(Q)$ of center $Q^{\text{ref}}$ and curvature $\kappa$ to the Hamiltonian of the unconstrained liquid. The factor $N$ ensures that the Hamiltonian remains an extensive quantity. By increasing $Q^{\text{ref}}$, one can explore different regions of the phase space that are characterized by larger overlap values, while the strength of the bias $\kappa$ mostly controls the amplitude of the fluctuations of $\widehat{Q}$.

In principle, a source $\epsilon$ could be directly applied to force the system toward large values of the overlap. However, the system is expected to slow down close to the putative critical point or near phase coexistence because of an increase in the extent of the overlap fluctuations. As discussed in Sec. V and in Ref. [57], the dynamics close to the random-field-like critical point is known to be activated with the relaxation time scaling exponentially (and not algebraically) with the variance of the order parameter. In addition, near the first-order transition line, the dynamics is dominated by rare nucleation events from the low-overlap phase to the high-overlap one. Overall, the direct study of the constrained liquid with a nonzero $\epsilon$ may thus give rise to severe sampling issues [17–19], even with the hybrid MD/swap scheme. By contrast, umbrella sampling enables one to control the amplitude of the overlap fluctuations and to make them small-enough to be accurately sampled.

For a given reference configuration $\boldsymbol{r}_0^N$ and a temperature $T$, we thus run $n_{\text{s}} \in [23, 35]$ simulations in parallel with umbrella potentials $\{W_k\}_{k=1...n_{\text{s}}}$ of identical curvature $\kappa$ and increasing centers $\{Q_k^{\text{ref}}\}_{k=1...n_{\text{s}}}$ in order to sample the entire range of overlap values between 0 and 1: see Fig. 15. In the $2d$ system, we use systems of moderate size (typically, up to $N = 250$) and simulations are very slow when a large bias strength $\kappa$ is imposed, as found in past work [129]. Consequently, we choose a smaller value of the bias strength $\kappa$ (see Table I) which results in a significant overlap between adjacent biased distributions of the overlap (see Sec. B 3). In $3d$ instead, we consider unprecedently large system sizes (typically, up to $N = 2400$) for such a type of simulation to perform a finite-size scaling analysis. In order for our reweighting scheme to adequately scale with $N$, we then use a large

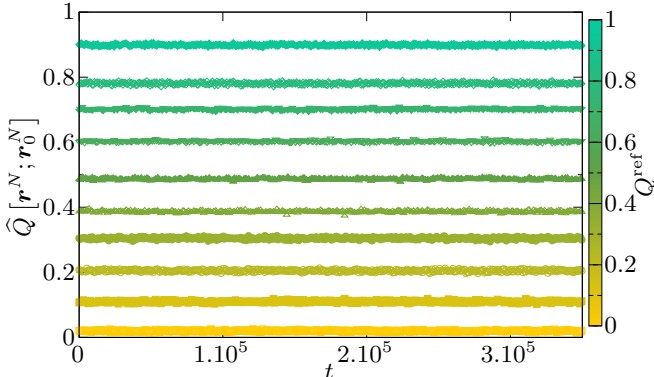

FIG. 15. Overlap time series for several biases $Q^{\mathrm{ref}}$ for the $3d$ liquid with $N = 1200$, $T = 0.22$, and the other parameters given in Table I.

bias strength $\kappa$ (see Table I) to reduce the fluctuations (see Sec. B 4).

For each biased simulation, the system is first equilibrated for $-t_{\mathrm{relax}} < t < 0$. Equilibration is ensured by checking that simulations started from two distinct initial conditions converge toward the same stationary state [36, 130]. Then, the statistical properties of the overlap are measured for $0 < t < t_{\mathrm{eq}}$. In $3d$, we monitor the mean-squared displacement,

$$\Delta(t) = \frac{1}{N} \sum_{i=1}^{N} |\boldsymbol{r}_i(t) - \boldsymbol{r}_i(0)|^2, \qquad (B4)$$

and we check that at the end of sampling, it exceeds a target value of 10. In $2d$, due to the so-called Mermin-Wagner fluctuations that induce large, and somehow spurious, translational displacements [76, 77], we instead follow the time evolution of the bond-orientational correlation function and require that it has decreased to 0 at the end of the sampling. The bond-orientational correlation function is defined as

$$C_{\psi_6}(t) = \frac{1}{N} \sum_{j=1}^{N} \psi_6^{(j)}(t) \left[ \psi_6^{(j)}(0) \right]^*, \qquad (B5)$$

where the star denotes the complex conjugate and

$$\psi_6^{(j)}(t) = \frac{1}{n_j(t)} \sum_{l=1}^{n_j(t)} e^{i6\theta_{jl}(t)}. \qquad (B6)$$

In the above equation, $n_j(t)$ is the number of neighbors of particle $j$ at time $t$, which are particles $l$ fulfilling the condition $|\boldsymbol{r}_j(t) - \boldsymbol{r}_l(t)|/\sigma_{jl} < 1.33$, and $\theta_{jl}(t)$ is the angle between the $x$-axis and the line joining the centers of the two neighbors [63]. As this correlation is rotationally invariant, the choice of the $x$-axis is made without any loss of generality. These criteria ensure that particles in both $2d$ and $3d$ have moved sufficiently and that the system explores the phase space ergodically.

### 3. Multi-histogram reweighting

In $2d$, we use a method already used in Refs. [26, 24, 40, 41] to compute $\mathcal{P}(Q; \boldsymbol{r}_0^N)$ from the different biased simulations. It relies on the Weighted Histogram Analysis Method (WHAM) [131, 132] which is an extension to arbitrary collective variables (such as the overlap) and potential biases of the multiple histogram method [73, 133, 134] first developed with the aim of extrapolating the thermodynamic properties of the Ising model at temperatures at which the system was not directly simulated.

We give a derivation of the formula that allows us to reconstruct $\mathcal{P}(Q; \boldsymbol{r}_0^N)$ from the $n_{\mathrm{s}}$ simulations run with the different biases. For the $k^{\mathrm{th}}$ simulation run at a temperature $T$ with a reference configuration $\boldsymbol{r}_0^N$, the empirical histogram of the overlap is

$$\frac{\mathcal{N}_k(Q)}{n_k} = \frac{1}{\mathcal{Z}_k} \mathcal{P}(Q; \boldsymbol{r}_0^N) e^{-\beta W_k(Q)}, \qquad (B7)$$

where $n_k$ is the total number of times the overlap was stored during the $k^{\mathrm{th}}$ simulation and $\mathcal{Z}_k$ is a normalization constant. In consequence, from one biased histogram, it is in principle possible to determine the unconstrained probability distribution of the overlap by inverting the above equation. However, during a simulation of finite duration $t_{\mathrm{eq}}$, only a restricted range of overlap values is sampled and, in practice, we can only use the above equation to determine $\mathcal{P}(Q; \boldsymbol{r}_0^N)$ in the range in which the histogram has nonzero values. However, as is clearly visible from Fig. 15, this range changes from one simulation to the other, and we thus seek $\mathcal{P}(Q; \boldsymbol{r}_0^N)$ for the entire range $[0, 1]$ as a linear combination of its estimate from each separate biased histogram, i.e.,

$$\mathcal{P}(Q; \boldsymbol{r}_0^N) = \sum_{k=1}^{n_{\mathrm{s}}} y_k n_k^{-1} \mathcal{Z}_k \mathcal{N}_k(Q) e^{\beta W_k(Q)}, \qquad (B8)$$

where $\{y_k\}_{k=1...n_{\mathrm{s}}}$ are unknown coefficients that verify the condition

$$\sum_{k=1}^{n_{\mathrm{s}}} y_k = 1. \qquad (B9)$$

To determine the coefficients $y_k$, we require that the statistical error on the above estimate is minimum. The histograms for the different biased simulations are independently measured, and the squared statistical error on $\mathcal{P}(Q; \boldsymbol{r}_0^N)$ reads:

$$[\delta \mathcal{P}(Q; \boldsymbol{r}_0^N)]^2 = \sum_{k=1}^{n_{\mathrm{s}}} y_k^2 n_k^{-2} \mathcal{Z}_k^2 [\delta \mathcal{N}_k(Q)]^2 e^{2\beta W_k(Q)}. \qquad (B10)$$

To estimate the statistical error on the biased histogram $\mathcal{N}_k(Q)$, we make a thought experiment. We assume that we have performed $n_h$ times the same simulation with the same bin center $Q_k^{\mathrm{ref}}$ during which we have

measured $n_k$ times the value of the overlap. For instance, this would correspond to simulations with different initial conditions or different sequences of random numbers for swap moves. Then, for each bin, the statistical error is given by the variance computed over the $n_h$ histograms. If we let brackets $\langle\langle\cdot\rangle\rangle$ denote the average over the $n_h$ simulations, the statistical error on the biased histogram is given by [135]

$$[\delta\mathcal{N}_k(Q)]^2 = g_k\langle\langle\mathcal{N}_k(Q)\rangle\rangle\left\{1 - \frac{\langle\langle\mathcal{N}_k(Q)\rangle\rangle}{n_k}\right\}, \quad \text{(B11)}$$

where $g_k$ is the statistical inefficiency, which is given by $g_k = 1 + 2\tau_k/\Delta t_k$ with $\tau_k$ the (auto)correlation time of the overlap for the $k^{\text{th}}$ simulation and $\Delta t_k(= \mathrm{d}t)$ the time interval between two measures of the overlap. If the bin width is small-enough, or if the overlap range that is covered during the $k^{\text{th}}$ simulation is sufficiently large, then $\langle\langle\mathcal{N}_k(Q)\rangle\rangle \ll n_k$ and [73, 135]:

$$\begin{aligned}[\delta\mathcal{N}_k(Q)]^2 &\approx g_k\langle\langle\mathcal{N}_k(Q)\rangle\rangle \\ &= n_k g_k \mathcal{Z}_k^{-1}\mathcal{P}(Q;\boldsymbol{r}_0^N)e^{-\beta W_k(Q)}.\end{aligned} \quad \text{(B12)}$$

Eventually, one obtains for the statistical error on the unconstrained probability distribution of the overlap

$$\begin{aligned}&\left[\delta\mathcal{P}(Q;\boldsymbol{r}_0^N)\right]^2 \\ &= \mathcal{P}(Q;\boldsymbol{r}_0^N)\sum_{k=1}^{n_{\mathrm{s}}} y_k^2 n_k^{-1}g_k\mathcal{Z}_k e^{N\beta W_k(Q)}.\end{aligned} \quad \text{(B13)}$$

To minimize the previous expression with respect to the $y_k$'s with the constraint given by Eq. (B9), we introduce the Lagrangian

$$\mathcal{L} = [\delta\mathcal{P}(Q;\boldsymbol{r}_0^N)]^2 - \varsigma\sum_{k=1}^{n_{\mathrm{s}}}y_k, \quad \text{(B14)}$$

with $\varsigma$ a Lagrange multiplier. The coefficients $y_k$ are thus given by $\partial\mathcal{L}/\partial y_k = 0$, which yield

$$y_k = \frac{\varsigma}{2\mathcal{P}(Q;\boldsymbol{r}_0^N)}n_k g_k^{-1}\mathcal{Z}_k^{-1}e^{-\beta W_k(Q)}, \quad \text{(B15)}$$

and using again Eq. (B9) to determine the Lagrange multiplier, we finally obtain

$$\mathcal{P}(Q;\boldsymbol{r}_0^N) = \frac{\displaystyle\sum_{k=1}^{n_{\mathrm{s}}} g_k^{-1}\mathcal{N}_k(Q)}{\displaystyle\sum_{k=1}^{n_{\mathrm{s}}} n_k g_k^{-1}\mathcal{Z}_k^{-1}e^{-\beta W_k(Q)}}. \quad \text{(B16)}$$

Once the partition functions are known, the unconstrained probability distribution of the overlap can then be determined. The partition functions can be expressed by using Eq. (B7), summing over all bins and inserting the previous equation:

$$\mathcal{Z}_k = \int_0^1 \mathrm{d}Q\,\frac{\displaystyle\sum_{k'=1}^{n_{\mathrm{s}}} g_{k'}^{-1}\mathcal{N}_{k'}(Q)}{\displaystyle\sum_{k'=1}^{n_{\mathrm{s}}} n_{k'}g_{k'}^{-1}\mathcal{Z}_{k'}^{-1}e^{-N\beta[W_{k'}(Q)-W_k(Q)]}}. \quad \text{(B17)}$$

We have checked that the statistical inefficiencies are not varying much from one biased simulation to another, and we can simplify the previous equations by setting $g_k = 1$ for all $k$.

The set of equations (B17) is solved self-consistently starting from $\mathcal{Z}_k = 1$ for all $k$. The iteration is stopped when the relative change in the partition function between two iterations is less than $10^{-10}$. To avoid overflows or underflows, the partition functions are rescaled at each iteration by the geometric average of the minimum and the maximum partition function over all the simulations. In practice, the convergence of the partition functions is fast and the result of the reweighting procedure only weakly depends on the cut-off criterion to stop the iteration [73]. Once the partition functions are converged, the probability distribution can be readily obtained from Eq. (B16). We emphasize that, with this procedure, we are able to determine $\mathcal{P}(Q;\boldsymbol{r}_0^N)$ on the full range $[0, 1]$, hence to measure exponentially small values in $N$ of the overlap probability distribution.

The accuracy of the reweighting procedure using WHAM requires a significant overlap between adjacent histograms. As the width of the histograms is expected to shrink with $N$ as $1/\sqrt{N}$, increasing the system size requires a larger number of simulations. We could also decrease the bias curvature $\kappa$ but this would be problematic as this also decreases the driving force toward configurations with untypically large overlap values. In $2d$, with the moderate sizes that we consider, the multi-histogram method is suitable. In $3d$, we consider larger system sizes up to $N = 2400$. We thus turn to another reweighting procedure. It is similar to the umbrella integration [136] or the Gaussian ensemble [72, 137], and does not require a significant overlap between adjacent distributions.

### 4. Gaussian ensemble reweighting

In $3d$, instead of setting $\kappa$ to a small value to have adjacent overlapping biased histograms, we apply a bias with a large curvature $\kappa$ in order for the biased histograms to display a sharp peak at their most probable value which we denote by $Q_k^*$ for $k = 1\ldots n_{\mathrm{s}}$. Taking the logarithm of Eq. (B7), differentiating with respect to $Q$, and evaluating at the most probable value yield:

$$V'(Q_k^*;\boldsymbol{r}_0^N) = -\frac{W_k'(Q_k^*)}{N} = \kappa\left(Q_k^{\mathrm{ref}} - Q_k^*\right), \quad \text{(B18)}$$

where the prime denotes a derivative with respect to $Q$ and $V(Q;\boldsymbol{r}_0^N)$ is the large deviation rate function of

$\mathcal{P}(Q; \boldsymbol{r}_0^N)$, namely, the random Franz-Parisi potential,

$$\mathcal{P}(Q; \boldsymbol{r}_0^N) \propto e^{-N\beta V(Q; \boldsymbol{r}_0^N)}. \tag{B19}$$

We note at this point that the normalization constants $\mathcal{Z}_k$ have disappeared from the expression of the bulk probability distribution (or equivalently its large deviation rate function). Consequently, for each simulation, we just need to measure the most probable value of the overlap. We end up with $n_s$ values of the derivative of $V(Q; \boldsymbol{r}_0^N)$ estimated at $n_s$ different points. As $Q^{\mathrm{ref}}(Q^*)$ is a smooth function, we interpolate it by means of a cubic spline [138]. Finally, the cubic spline can be analytically integrated to obtain $V(Q; \boldsymbol{r}_0^N)$ up to an additive constant which we choose so that $V(Q; \boldsymbol{r}_0^N)$ is zero at its global minimum:

$$V(Q; \boldsymbol{r}_0^N) = \kappa \int_{Q_{\mathrm{rand}}}^{Q} Q^{\mathrm{ref}}(Q^*)\mathrm{d}Q^* - \frac{1}{2}\kappa\left(Q^2 - Q_{\mathrm{rand}}^2\right), \tag{B20}$$

with $Q^{\mathrm{ref}}(Q^*)$ locally approximated by a third-degree polynomial function [139]. The full procedure is represented in Fig. 16(a). The probability distribution is eventually obtained from Eq. (B19): see Fig. 16(b). Once again, we stress that, with this procedure, we are able to sample the large deviation rate function associated with $\mathcal{P}(Q; \boldsymbol{r}_0^N)$ on the full range of overlap values and, as a result, to measure arbitrary small probabilities in $N$ (less than $10^{-300}$).

We now explain how to determine the most probable value of the overlap for a given biased simulation during the course of the simulation, without actually measuring the histogram $\mathcal{N}_k(Q)$, to avoid systematic errors related to the bin width. Our goal is to derive an expression for the most probable value from quantities directly accessible during a simulation, such as the cumulants of the overlap. To obtain more insight about this relation we show in Fig. 16(c) the skewness

$$\gamma_k^{(1)} = \frac{\langle(\widehat{Q} - \langle\widehat{Q}\rangle_k)^3\rangle_k}{\langle(\widehat{Q} - \langle\widehat{Q}\rangle_k)^2\rangle_k^{3/2}} \tag{B21}$$

and the kurtosis

$$\gamma_k^{(2)} = \frac{\langle(\widehat{Q} - \langle\widehat{Q}\rangle_k)^4\rangle_k}{\langle(\widehat{Q} - \langle\widehat{Q}\rangle_k)^2\rangle_k^2} - 3, \tag{B22}$$

where $\langle\cdot\rangle_k$ denotes the thermal average in the $k^{\mathrm{th}}$ simulation. They are both close to 0, which is their expected value if the overlap is normally distributed. Besides, the kurtosis remains small for all biases while the skewness is larger for extreme values of $Q^{\mathrm{ref}}$. Therefore, it is reasonable to assume that the biased histograms $\mathcal{N}_k(Q)$ are well approximated by [72]

$$\mathcal{N}_k(Q) \propto e^{-\alpha_k(Q-Q_k^*)^2 + \xi_k(Q-Q_k^*)^3} \\ \propto \left[1 + \xi_k(Q-Q_k^*)^3\right]e^{-\alpha_k(Q-Q_k^*)^2}, \tag{B23}$$

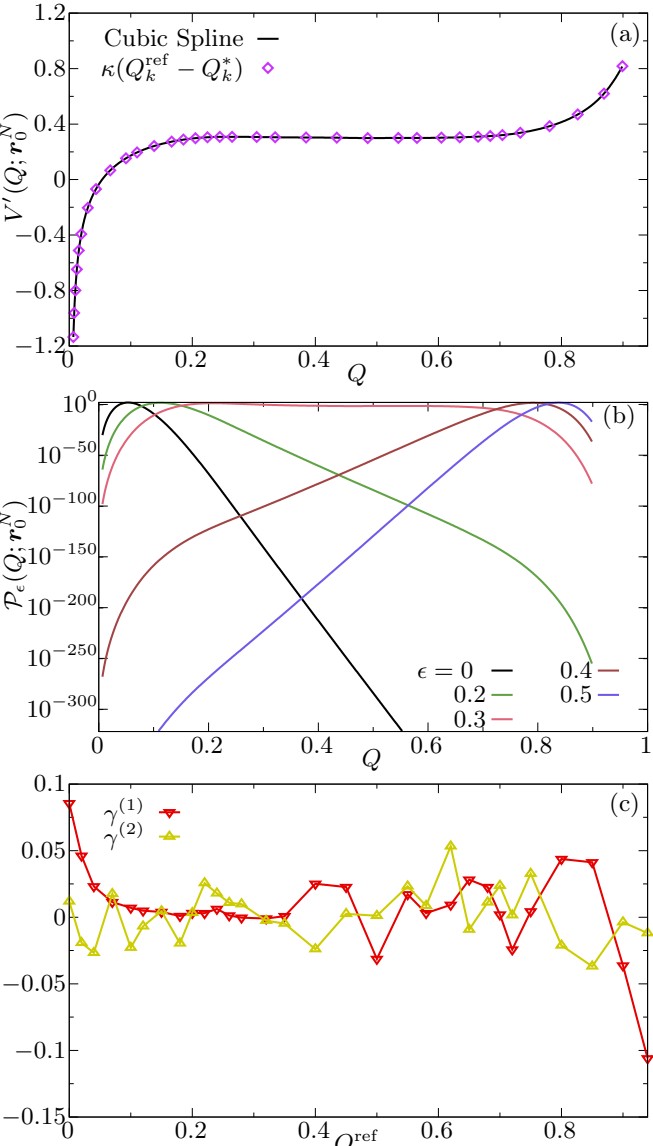

FIG. 16. Gaussian ensemble reweighting for the 3$d$ liquid with $N = 1200$, $T = 0.22$, and the other parameters given in Table I. (a) Derivative of the large deviation rate function $V(Q; \boldsymbol{r}_0^N)$ at the discrete most probable values $\{Q_k^*\}_{k=1...n_s}$ and its cubic spline interpolation. (b) Unconstrained probability distribution $\mathcal{P}(Q; \boldsymbol{r}_0^N)$ ($\epsilon = 0$) obtained by integration of the cubic spline and by using Eq. (B19), along with distributions $\mathcal{P}_\epsilon(Q; \boldsymbol{r}_0^N) \propto \mathcal{P}(Q; \boldsymbol{r}_0^N)e^{N\beta\epsilon Q}$ of the overlap for finite values of $\epsilon = 0.2, 0.3, 0.4, 0.5$. (c) Skewness $\gamma_k^{(1)}$ [see Eq. (B21)] and kurtosis $\gamma_k^{(2)}$ [see Eq. (B22)] of the biased histograms $\mathcal{N}_k(Q)$ as a function of the bias center $Q^{\mathrm{ref}}$.

where the third-order term is considered as a perturbation of the Gaussian limit ($\xi_k = 0$) and is nonzero for extreme values of $Q^{\mathrm{ref}}$ only. We restrict ourselves to expansions at the first order in $\xi_k$, which are correct if $\xi_k\alpha_k^{-3/2} \ll 1$. Expansions at any order could be done but this requires measuring an increasing number of cumulants of the overlap in each biased simulation, which

may give rise to larger statistical errors if $t_{\text{eq}}$ is not large-enough.

We use Eq. (B23) to compute the three first cumulants of the overlap, which then read at the leading order in $\xi_k$

$$\langle \widehat{Q} \rangle_k = Q_k^* + \frac{3\xi_k}{4\alpha_k^2},$$

$$\langle (\widehat{Q} - \langle \widehat{Q} \rangle_k)^2 \rangle_k = \frac{1}{2\alpha_k}, \qquad (B24)$$

$$\langle (\widehat{Q} - \langle \widehat{Q} \rangle_k)^3 \rangle_k = \frac{3\xi_k}{4\alpha_k^3}.$$

Inserting the second and third lines of Eq. (B24) in the first one leads to

$$Q_k^* = \langle \widehat{Q} \rangle_k - \frac{\langle (\widehat{Q} - \langle \widehat{Q} \rangle_k)^3 \rangle_k}{2\langle (\widehat{Q} - \langle \widehat{Q} \rangle_k)^2 \rangle_k}. \qquad (B25)$$

The right-hand side can be measured on the fly in simulations and the most probable value of the overlap can be obtained from the measured moments of the overlap. The small parameter involved in the previous expansions, $\xi_k \alpha_k^{-3/2} = \sqrt{2}\gamma_k^{(1)}$, is directly related to the skewness of the biased histogram. Fig. 16(c) shows that this parameter is indeed much smaller than 1, making our approach fully self-consistent. We also note that if the biased histogram is symmetric and almost Gaussian the above expression reduces to $Q_k^* = \langle \widehat{Q} \rangle_k$. Inserting this into Eq. (B18) yields the reweighting formula for a related interpolation scheme known as the tethered Monte Carlo method [140] which has already been implemented in the context of supercooled liquids and glasses [27, 86]. However, the method used in the present study has the merit of being able to cure the zeroth-order Gaussian approximation of the tethered method by storing an increasing number of cumulants of the overlap order parameter during umbrella simulations. (Of course the measurement of higher-order cumulants of the overlap would require longer simulations.)

The Gaussian approximation is even more accurate when $\kappa$ is large. However, if $\kappa$ becomes too large, the amplitude of the bias force applied on each particle grows and the time step for integrating the equations of motion must be decreased to keep the same numerical accuracy and continue to sample the phase space correctly. A trade off is thus necessary.

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
