# Peer review of "Statistical mechanics of coupled supercooled liquids in finite dimensions"

_SciPost Physics_

## Round 1 · Referee Report · Anonymous (Referee 1) · 2021-7-1

Strengths

1) The paper is interesting, clear and well written. 2) It combines numerical and analytical work, showing a remarkably similar behavior. 3) The numerical data shown are of high quality. Authors manage to thermalize unprecedentedly large system sizes through the combination of biased sampling methods and the swap algorithm.

Weaknesses

1) Most of the results discussed in the paper are not new, and the text does not properly reflect this fact, in my opinion.

Report

The paper by Guiseling et al. studies a very interesting problem in glasses, -- the statistical mechanics of two coupled replicas of supercooled liquids--, that has important implications for the understanding of the physical mechanisms beneath the glass transition. The paper approaches the problem from two complementary points of view: (i) analytically, through the p-spin model, and (ii) numerically, via simulations of a soft particle model. The manuscript is well written and describes in detail both the results obtained, but also the methods used and important points in the problem, such as the thermalization protocol.

In the way the paper is written, and especially the abstract, it makes the reader think that the results are more groundbreaking than they are. This is a problem that has been studied before, the RFIM universality class had already been tested in simulations and so were the statistical mechanics of the p-spin model. Yet, this paper presents new calculations obtained when varying the temperature of the reference configuration T_0, that seemed crucial to achieve thermalization of unprecedentedly large supercooled liquid configurations even at large values of the constrained overlaps. Such simulations have allowed to, first, suggest that the epsilon line disappears in 2d, and second, prove without any doubt, the RFIM nature of the critical point. From this last comment, I think the paper merits publication, but before being published in SciPost, authors should make clearer what are the real progresses of this work.

Requested changes

1) The p-spin is not defined in the main-text, and in the present form, the quantities shown in figure 2, such as J and Delta are not defined. 2) I am a bit confused about the discussion about the absence of interface free energy cost below figure 3. The strictly increasing nature of the isotherms in Fig. 3) would not disappear if the same curve were represented against the constraint Q? The probabilities in Fig 4. do show an interfacial free energy between the peaks. 4) The disappearance of the 2d transition is not properly studied. As far as I understand, the authors just show that the critical temperature is moving towards a lower temperature with the system size, but no proper finite size analysis have been used. Authors could attempt to compute the correlation length, xi, and show the behavior of xi/L with T. 5) The combination of the Gaussian ensemble reweighing with the umbrella sampling method used for 3d results is the Tethered method [Martin-Mayor, V., Seoane, B., & Yllanes, D. (2011). Tethered Monte Carlo: Managing rugged free-energy landscapes with a Helmholtz-potential formalism. Journal of Statistical Physics, 144(3), 554-596.] and it was previously used to compute the Franz-Parisi potential in the Ref. [25] of this work and [Cammarota, C., & Seoane, B. (2016). First-principles computation of random-pinning glass transition, glass cooperative length scales, and numerical comparisons. Physical Review B, 94(18), 180201. ].

  • validity: high
  • significance: high
  • originality: good
  • clarity: top
  • formatting: excellent
  • grammar: excellent

Author:  Benjamin Guiselin  on 2021-11-29  [id 1985]

(in reply to Report 1 on 2021-07-01)

The paper by Guiselin et al. studies a very interesting problem in glasses, -- the statistical mechanics of two coupled replicas of supercooled liquids--, that has important implications for the understanding of the physical mechanisms beneath the glass transition. The paper approaches the problem from two complementary points of view: (i) analytically, through the p-spin model, and (ii) numerically, via simulations of a soft particle model. The manuscript is well written and describes in detail both the results obtained, but also the methods used and important points in the problem, such as the thermalization protocol.

We thank the Referee for his/her careful reading and positive appreciation of our work. We address his/her comments and detail the changes made in the manuscript below. He/She provided a number of suggestions to improve the manuscript, which we mostly followed.

In the way the paper is written, and especially the abstract, it makes the reader think that the results are more groundbreaking than they are. This is a problem that has been studied before, the RFIM universality class had already been tested in simulations and so were the statistical mechanics of the p-spin model. Yet, this paper presents new calculations obtained when varying the temperature of the reference configuration T_0, that seemed crucial to achieve thermalization of unprecedentedly large supercooled liquid configurations even at large values of the constrained overlaps. Such simulations have allowed to, first, suggest that the epsilon line disappears in 2d, and second, prove without any doubt, the RFIM nature of the critical point. From this last comment, I think the paper merits publication, but before being published in SciPost, authors should make clearer what are the real progresses of this work.

We agree that the statistical mechanics of coupled replicas of glass-forming liquids has been studied before, that the RFIM universality class has already been tested in simulations and the statistical mechanics of the $p$-spin model been investigated before. To acknowledge this, we had referred to 50-60 prior publications, and it was not at all our intention to oversell our own study. The abstract is a genuine description of our work (and of the key role played by the use of a low reference temperature $T_0$) with no intention to hide previous work. However, to avoid any misinterpretation, we have toned down what could have been read as overhyping our results. We have also rewritten the last 3 sentences of the abstract and added a sentence at the end of Section III with even a few more references. We have modified the last 3 paragraphs of the Introduction to clarify what we achieve and what is new in our contribution to the problem, and rewritten part of the Conclusion.

1) The p-spin is not defined in the main-text, and in the present form, the quantities shown in figure 2, such as J and Delta are not defined.

As suggested by the Referee, we have added the definition of the $p$-spin model in the main text [Eq. (4) and below] and also modified the caption of Fig. 2.

2) I am a bit confused about the discussion about the absence of interface free energy cost below figure 3. The strictly increasing nature of the isotherms in Fig. 3) would not disappear if the same curve were represented against the constraint Q? The probabilities in Fig 4. do show an interfacial free energy between the peaks.

Nowhere in this section do we write that there is no interfacial free-energy cost. The Referee is right that in the systems shown in Fig. 4 at low enough temperature there is an interfacial free-energy cost between low- and high-overlap phases which is related to the minimum of the probability distribution at coexistence.

The isotherms in Fig. 3 are always increasing because, as explained in the text, in the canonical ensemble (fixed $\epsilon$) we compute the first cumulant of the overlap probability distribution, which is therefore an average over the whole $Q$-range including low- and high-$Q$ regions. The resulting isotherm has to be monotonically increasing, $\textit{i.e.}$, for a given value of $\epsilon$ there is a unique value of the first cumulant. This is not in contradiction with the existence of a minimum in the overlap probability distribution. If instead one represents the average field $\kappa(Q^\mathrm{ref}-\langle\widehat Q\rangle)$ as a function of the average overlap $\langle\widehat Q\rangle$ in umbrella simulations [see Fig. 15(a)], one obtains a non-monotonic curve at low-enough temperatures, as found for instance with the Tethered Monte Carlo method that the Referee mentions below.

We want to stress that this difference in behavior in the first-order transition region between the umbrella simulations (Gaussian ensemble) and the results in the presence of a field $\epsilon$ (canonical ensemble) is a finite-size effect and that the two ensembles must be equivalent in the thermodynamic limit. In the Gaussian ensemble, the isotherms become less and less non-monotonic while in the canonical ensemble the slope at the inflexion point decreases when the size of the system is increased, until the two curves coincide in the thermodynamic limit with a flat isotherm at coexistence, corresponding to the Maxwell construction.

We have modified the corresponding paragraph to make it clearer.

4) The disappearance of the 2d transition is not properly studied. As far as I understand, the authors just show that the critical temperature is moving towards a lower temperature with the system size, but no proper finite size analysis have been used. Authors could attempt to compute the correlation length, xi, and show the behavior of xi/L with T.

Our manuscript was not clearly written as far as the two-dimensional case is concerned. Contrary to the $3d$ case for which we do perform a finite-size scaling analysis, strictly speaking we do not attempt to do the same for the $2d$ case. The reason is that our goal is to determine whether or not a signature of the transition exists at the lowest accessible temperatures. We have found that for all the temperatures that we can access, which cover a range that goes below the extrapolated calorimetric glass transition temperature $T_g$, the disorder-averaged probability distribution of the overlap starts bimodal for small system sizes but becomes unimodal for a size $N=250$. No plausible scenario would predict that the probability distribution could become bimodal again. This represents therefore very strong evidence that no phase transition takes place, without any need for a finite-size scaling analysis. A finite-size scaling analysis in $2d$, which is the expected lower critical dimension from the connection to the RFIM, would anyhow require much larger system sizes, far beyond what is achievable in simulations even with all the tricks that we use, because of the peculiar nature of the scaling: for instance, in studies of the $2d$ RFIM (see [Meinke-Middleton,arXiv:cond-mat/0502471], [Seppala-Alava, PRE 2001], or [Raju et al., PRX 2019]), systems of up to $10^6$ or more spins (studied at zero temperature) are considered, allowing the authors to properly assess the exponentially growing behavior of the correlation length with decreasing disorder and other features of scaling at a lower critical dimension. This is clearly out of reach for constrained glass-forming liquids, especially at the very low temperatures that we are considering. In consequence, we have not attempted a $\textit{bona fide}$ finite-size scaling in $2d$.

In the new version of the manuscript, we have now clarified our treatment of the $2d$ systems (we have changed the title of Section IV and rewritten subsections IV-A and IV-C). We give strong evidence that there is no phase transition in $d=2$ in the range of temperature that goes below the calorimetric glass transition temperature but strictly speaking, as we do not perform a finite-size scaling analysis, we cannot prove that $d=2$ is the lower critical dimension: in principle, it could be between 2 and 3, although this would represent an awkward scenario at odds with what is anticipated.

In addition, we have added new data for the $2d$ case (see below the response to Referee 2) in a new subsection IV-D.

5) The combination of the Gaussian ensemble reweighing with the umbrella sampling method used for 3d results is the Tethered method [Martin-Mayor, V., Seoane, B., & Yllanes, D. (2011). Tethered Monte Carlo: Managing rugged free-energy landscapes with a Helmholtz-potential formalism. Journal of Statistical Physics, 144(3), 554-596.] and it was previously used to compute the Franz-Parisi potential in the Ref. [25] of this work and [Cammarota, C., & Seoane, B. (2016). First-principles computation of random-pinning glass transition, glass cooperative length scales, and numerical comparisons. Physical Review B, 94(18), 180201. ].

We thank the Referee for noting this point and for the references that we have added to the revised version of the manuscript. There is however a difference between our scheme and the usual tethered Monte Carlo method. In the tethered method, Eq. (B18) of the manuscript holds with the most probable value replaced by the average overlap in the umbrella simulation, which amounts to a softening of the constraint on the overlap in the definition of the Franz-Parisi potential. This approximation becomes less verified in small systems or systems with a small curvature of the confining umbrella potential. Instead, in our method, we do not change the definition of the Franz-Parisi potential but approximate the distribution of the overlap in the presence of the umbrella potential by a slightly skewed Gaussian distribution. This is the same level of approximation as for the tethered method which becomes more accurate when the size of the system is increased, by virtue of the central limit theorem. However, one advantage of our method is that our scheme could be made more accurate even for smaller systems by storing an increasing number of cumulants of the overlap order parameter during umbrella simulations (of course this would require longer simulations), contrary to the usual tethered method.

Anonymous on 2022-01-05  [id 2068]

(in reply to Benjamin Guiselin on 2021-11-29 [id 1985])
Category:
validation or rederivation

After carefully reading the authors' replies and the changes introduced in the manuscript, I must say that I am very satisfied with the answers, so my recommendation is to publish the article in its current form.

---

## Round 1 · Referee Report · Anonymous (Referee 2) · 2021-7-22

Strengths

1- Numerical simulations seem accurate, although not complete enough 2- Data analysis looks well done, although partial 3- The analytic solution of the spherical p-spin model looks valid, although not well explained

Weaknesses

1- Most of the results were already published 2- The authors do not state clearly which are the novel results in this manuscript 3- More data or a better analysis is needed to go beyond the statements made in previous works 4- The analytical solution of the p-spin model is not well explained and fails in making a useful connection to finite-dimensional models

Report

This manuscript is the "long version" of Ref. [55] by the same authors. I have nothing against publishing first a "short version" and then a long one if the latter is a substantial improvement with respect to the former. In the present case, it does not look so. The main results presented in this manuscript already appeared in Ref.[55], including most of the figures for 3d data. Even the novel parts are not completely satisfying: - 2d data are very scarce and the analysis should be improved; - the analytic solution to the p-spin model, relegated to Appendix A, is not easy to understand and the connection to a finite-dimensional model is not well explained. So, in my opinion, this manuscript needs to be substantially improved before an editorial decision can be taken.

Let me start commenting on a point that is partially discussed by the authors, but somehow overlooked in my opinion: the case $T=T_0$. The authors say this would be the right computation to do, but then say the numerical study is more demanding. "More demanding" is not impossible, and given that the authors should go beyond previously published results, I believe that showing and analyzing in detail this case would be an important point to improve this manuscript. By the way, given that the swap algorithm can thermalize down to T=0.055, I do not see huge numerical limitations in performing a detailed study with $T=T_0\in[0.06,0.08]$. In Ref.[55] only 2 sizes N=300 and 1200 were shown at T=0.085, so covering the suggested temperature range seems a reasonable way to improve over previous publications.

Moreover, the authors claim to have data for 2d systems and $T=T_0$ that would confirm the claim that a transition is absent in that case. Showing these data is strongly suggested, in order to make claims in the 2d case more robust.

The analysis of the 2d data is not satisfactory. How can the authors make any claim on the extrapolation to the thermodynamic limit from the 3 points shown in Figure 8? The authors should present for the 2d case at least the same information contained in Figure 7 for the 3d case. They should also check for any relation between connected and disconnected susceptibilities. Moreover, the data quality and quantity in the 2d case are too low with respect to the 3d case (and being the former the original part of this work the authors are urged to make it of comparable quality). I believe the authors should collect more data, similarly to the 3d case (where they use 7 temperatures and 4 sizes), and make a scaling analysis also for the 2d case. The final results must be quoted as an upper bound on the possible critical temperature. Assessments on the lack of a phase transition based on few points not showing a clear divergence are not enough.

On the 3d data, I believe that estimating the critical temperature from a scaling collapse is not a good choice. Given that the authors are collecting data down to the putative critical temperature, I think a more standard scaling analysis should be performed. The critical temperature must be estimated via numerical methods that provide also statistical uncertainty. Also, the analysis of correlations between different critical exponents would be a nice improvement over previously published work.

Regarding the analysis of the p-spin model, there are some points that are hard to understand. In particular, the meaning of the colored curves in the upper panel of Figure 11 are obscure to me, and I was unable to find a comprehensible explanation in the text.

I am also finding it somehow difficult to understand the case $T_0>T_\text{cvx}$. Indeed in this case the first-order phase transition takes place only for very low temperatures, much lower than $T_0$. I find it weird that the metastable state that one uses to constrain the measure does not exist ($T_0>T_d$) and not even after tilting the Franz-Parisi potential ($T_0>T_\text{cvx}$). I would not be able to provide any physical meaning to such a construction. So, either the authors are able to give a physical meaning, otherwise, I am afraid the mathematical results obtained by solving saddle point equations do not have a corresponding physical meaning (and as such should be ignored).

More importantly, the results about the spherical p-spin model, as they are presented now, are not very useful, because the connection to finite-dimensional models is not sufficiently detailed. When considering nucleation processes, how the phase diagrams of Figure 12 are expected to be modified? In which situation the 2d model should fall?

In deriving the replicated effective action, the authors do not include mixed terms with both fields and field derivatives. Why?

Eq. (A33) is wrong on the second line. I believe it should be h.m

After deriving the effective action the authors disregard all the sources of fluctuations but those in the random external field. Obviously in this way, one gets a random field Ising model, by definition. But instead of assuming the other fluctuating terms are irrelevant, the authors should show this irrelevance.

Minor points: - in Figure 10 some squares that should be empty look as filled because of the line passing through it - I would complement Figure 12 with a further figure showing the 4 colored curves together, so as to better show how the critical line gets reduced increasing $T_0$

Requested changes

1- Clarify what is new in this manuscript and which results already appeared in previous publications 2- Discuss in more detail the case $T=T_0$ showing appropriate data 3- Improve the presentation of the analytical results for the p-spin model and discuss the connection with finite-dimensional models 4- Provide new data or a better analysis to go beyond the results already published

  • validity: top
  • significance: high
  • originality: ok
  • clarity: high
  • formatting: excellent
  • grammar: excellent

Author:  Benjamin Guiselin  on 2021-11-29  [id 1986]

(in reply to Report 2 on 2021-07-22)

We first would like to thank the Referee for his/her careful reading of the manuscript. We address his/her comments and detail the changes made in the manuscript below.

This manuscript is the "long version" of Ref. [55] by the same authors. I have nothing against publishing first a "short version" and then a long one if the latter is a substantial improvement with respect to the former. In the present case, it does not look so. The main results presented in this manuscript already appeared in Ref.[55], including most of the figures for 3d data. Even the novel parts are not completely satisfying: - 2d data are very scarce and the analysis should be improved; - the analytic solution to the $p$-spin model, relegated to Appendix A, is not easy to understand and the connection to a finite-dimensional model is not well explained. So, in my opinion, this manuscript needs to be substantially improved before an editorial decision can be taken.

We disagree with the Referee concerning what should be an appropriate long paper published after a short version. We believe it is important to first provide the scientific community with all the details concerning the methods, the treatments, and the models that are implemented to obtain the results. This was only sketched in the short report and this is what we do in the present manuscript, both in the main text and in the appendices. We have used sophisticated techniques to be able to optimize the numerical strategy and simulate as large as possible, as many as possible, as low-temperature as possible systems with present-day computer ressources, going much beyond what had been achieved before. It is then worth detailing the strategy and the techniques. We nonetheless agree with the Referee that a mere restatement of results already described in the short version would make a dull paper, but this is not what we do in our manuscript. The short version presented a finite-size scaling analysis as well as an investigation of the critical dynamics for the $3d$ liquid. The long version expands on the results for the $3d$ system (display of the evolution of the Franz-Parisi potential, finite-size study of the free-energy barrier between low- and high-overlap phases, illustration of phase separation on a configuration of 10000 particles, etc.) and presents new results: in particular, we explain the dependence on the temperature $T_0$ of the reference configurations via the $p$-spin model, a key point for our numerical strategy, and we provide simulations concerning the $2d$ system at very low temperatures below the extrapolated calorimetric glass transition temperature. The latter allows us to clearly contrast the behavior in $d=3$ and in $d=2$, as is strikingly illustrated by the evolution of the overlap probability distribution with increasing system size. Of course, one would always like to have more data, bigger system sizes, more temperatures, etc., but the numerical effort that our study represents is already enormous and it is not reasonable at this point to launch a full new investigation, whether in $d=2$, or $d=3$. We discuss in more detail the $2d$ case below.

Let me start commenting on a point that is partially discussed by the authors, but somehow overlooked in my opinion: the case $T=T_0$. The authors say this would be the right computation to do, but then say the numerical study is more demanding. "More demanding" is not impossible, and given that the authors should go beyond previously published results, I believe that showing and analyzing in detail this case would be an important point to improve this manuscript. By the way, given that the swap algorithm can thermalize down to T=0.055, I do not see huge numerical limitations in performing a detailed study with $T=T_0\in[0.06,0.08]$. In Ref.[55] only 2 sizes N=300 and 1200 were shown at T=0.085, so covering the suggested temperature range seems a reasonable way to improve over previous publications. Moreover, the authors claim to have data for 2d systems and T=T0 that would confirm the claim that a transition is absent in that case. Showing these data is strongly suggested, in order to make claims in the 2d case more robust.

A motto such as ``more demanding is not impossible'' sounds certainly nice as a motivational speech but after the huge simulation effort associated with the development of an efficient numerical strategy which we have achieved in our study we feel this is a somehow improper if not offensive comment. We have explained in the manuscript why the problem is not only to generate configurations at a low temperature and why it is much harder to study the case with $T=T_0$.

In passing, we do not say that $T=T_0$ ``would be the right computation to do''. The case $T=T_0$ corresponds to taking typical configurations at temperature $T$ to study glassy minima at this temperature and the putative random first-order transition (RFOT) for $\epsilon=0$ indeed takes place in $T=T_0=T_K$. However, the absence of a transition and of a critical point for a low $T_0$ fixed implies that there is no phase transition and critical point when $T=T_0$ (the critical temperature is expected to decrease as one increases $T_0$), hence that there is no RFOT. This seems to be what happens in $d=2$. In $d=3$, the critical point for $T=T_0$ is in the same universality class as the critical point for a low $T_0$ fixed. It is true that in this case, the existence of a transition for a low $T_0$ does not necessarily imply the existence of a transition when $T=T_0$.

We have nonetheless restructured the manuscript by adding a subsection IV-D in which we specifically discuss the case $T=T_0$. As suggested by the Referee, we present new data for the $2d$ case, together with data for the $3d$ case already shown in the short report. Showing the two sets of data together allows us to contrast the difference of behavior in $2d$ and $3d$. This confirms the absence of a transition in $2d$ and points to the fact that the transition survives in $3d$. Unlike for the other results the latter conclusion for the $3d$ case is not a strong proof, due to the absence of a systematic finite-size scaling analysis. We are careful in the manuscript about not making any unsubstantiated claims (and we have clarified the wording in the new version to stress the point).

We note that the distribution in $2d$ is not bimodal for $N=64$ and $T=0.12$ nor for $N=125$ and $T=0.06$ contrary to the case with a low $T_0=0.03$. This is fully consistent with the fact that the effective disorder is stronger for the case $T=T_0$, or equivalently that the linear size one must reach to see that the disorder destroys the transition decreases. This makes the study of overlap fluctuations more difficult in $2d$ for the case $T=T_0$ as it would also require simulations for system sizes smaller than 64 particles which is not achievable because of the tendency of the system to crystallize. This is one more reason why we mainly focused on the case of a low and fixed value of $T_0$.

The analysis of the 2d data is not satisfactory. How can the authors make any claim on the extrapolation to the thermodynamic limit from the 3 points shown in Figure 8? The authors should present for the 2d case at least the same information contained in Figure 7 for the 3d case. They should also check for any relation between connected and disconnected susceptibilities. Moreover, the data quality and quantity in the 2d case are too low with respect to the 3d case (and being the former the original part of this work the authors are urged to make it of comparable quality). I believe the authors should collect more data, similarly to the 3d case (where they use 7 temperatures and 4 sizes), and make a scaling analysis also for the 2d case. The final results must be quoted as an upper bound on the possible critical temperature. Assessments on the lack of a phase transition based on few points not showing a clear divergence are not enough.

The Referee seems to miss the point that it is easier in the present case to show the absence of a transition than to prove its existence. For the latter one really needs to carry out a finite-size scaling analysis with as large as possible system sizes. For the former, based on the accumulated experience on phase transitions and on the effect of fluctuations on critical points, it is sufficient to show that the trend to a bimodal order parameter distribution which is found in small systems disappears as one increases the system size: this signals that fluctuations extending over larger scales when the system size is larger destroy the phase transition; increasing the range of the fluctuations by taking even larger system sizes would make the disappearance even clearer (except if one envisages an odd scenario in which the phase transition makes a reappearance at a larger system size, but such a scenario would anyhow invalid any finite-size scaling analysis as new features could always make their entry at still larger system sizes than those under study, no matter how large they are).

Contrasting, as we do in this manuscript and is one of its novel aspects, the behavior in $2d$ and in $3d$ is particularly illuminating, and we have stressed even more this point in the new version. So, for our purpose, $\textit{i.e.}$, showing that there is no phase transition, there is no need for a finite-size scaling analysis. As we mentioned in the response to Referee 1, actually proving that $d=2$ is the lower critical dimension and that the correlation length grows exponentially when the temperature decreases would require huge sizes, probably of $10^6$ particles as done for the RFIM (see [Meinke-Middleton,arXiv:cond-mat/0502471], [Seppala-Alava, PRE 2001], or [Raju et al., PRX 2019]), sizes which are clearly out of reach for the present system. Again, the Referee does not seem to appreciate the amount of numerical work that our investigation represents. Of course, we encourage other groups to take up the challenge and push further the limits of what can be achieved for constrained glass-forming liquids.

We have added some new results and rewritten the Conclusion of our $2d$ study to clarify our goal and achievement. We also mention, as stated by the Referee, that we have only an upper bound on the critical temperature.

On the 3d data, I believe that estimating the critical temperature from a scaling collapse is not a good choice. Given that the authors are collecting data down to the putative critical temperature, I think a more standard scaling analysis should be performed. The critical temperature must be estimated via numerical methods that provide also statistical uncertainty. Also, the analysis of correlations between different critical exponents would be a nice improvement over previously published work.

As should be clear, it is out of reach to study very large system sizes that would allow us to carry out a full finite-size scaling analysis comparable to what is done for simpler critical systems or the RFIM. Furthermore, as we point out, the absence of a $Z_2$ inversion complicates the matter even more. So we cannot (and should not) push the analysis too far. We therefore use the known values of the critical exponents for the $3d$ RFIM to collapse data for several quantities. This provides an estimate of the critical temperature $T_c$. An uncertainty (although somehow arbitrary) on $T_c$ could be defined by imposing a criterion on the average quadratic difference between the curved built on the rescaled data and the master curve inferred from the algorithm we use. However, going beyond this with the available data does not seem sensible.

Regarding the analysis of the p-spin model, there are some points that are hard to understand. In particular, the meaning of the colored curves in the upper panel of Figure 11 are obscure to me, and I was unable to find a comprehensible explanation in the text. I am also finding it somehow difficult to understand the case $T_0>T_\mathrm{cvx}$. Indeed in this case the first-order phase transition takes place only for very low temperatures, much lower than $T_0$. I find it weird that the metastable state that one uses to constrain the measure does not exist ($T_0>T_d$) and not even after tilting the Franz-Parisi potential ($T_0>T_\mathrm{cvx}$). I would not be able to provide any physical meaning to such a construction. So, either the authors are able to give a physical meaning, otherwise, I am afraid the mathematical results obtained by solving saddle point equations do not have a corresponding physical meaning (and as such should be ignored).

Concerning the analysis of the $p$-spin model, we have followed the Referee's advice: we have improved the explanations and clarified the meaning of the curves in Fig. 11 (now Fig. 12). [The colored curves represent the locations of the low- and high-overlap minima of the tilted Franz-Parisi potential at coexistence, $\textit{i.e.}$, of $V(Q)-\epsilon Q$, in the first-order transition region when the latter displays two minima of equal depth.] We have also removed a figure and detailed explanations concerning the case with $T_0>T_\mathrm{cvx}$ whose physical implications beyond mean-field may indeed be questionable. We have also rewritten the penultimate paragraph of the subsection A-4 of the Appendix. Finally, we have clarified the discussion of the effective theory beyond mean-field in subsection A-5 (see the comments below).

More importantly, the results about the spherical p-spin model, as they are presented now, are not very useful, because the connection to finite-dimensional models is not sufficiently detailed. When considering nucleation processes, how the phase diagrams of Figure 12 are expected to be modified? In which situation the 2d model should fall?

We do not understand the Referee's comment about the relation between the mean-field $p$-spin model and finite-dimensional systems. This is a topic that has triggered dozens and dozens of papers and that remains vividly debated. What has been clarified is that the fully connected $p$-spin model is in the universality class of mean-field glass-formers, together with infinite-dimensional liquids. As we clearly detail in the Introduction, our goal is precisely to contribute to the assessment of what remains of the mean-field scenario of glass formation in finite-dimensional glass-forming liquids. We mention in particular that while nucleation processes wash away the mean-field dynamical (or mode-coupling) transition a line of first-order transition terminating in a critical point may still be present in the $(\epsilon,T)$ diagram.

Furthermore, while the fully-connected $p$-spin model is representative of the class of mean-field glass-formers, finite-dimensional $p$-spin models are not particularly interesting (see, $\textit{e.g.}$, [C. Cammarota, G. Biroli, M. Tarzia, and G. Tarjus, Phys. Rev. B 87, 064202 (2012)]). We just illustrate for the $p$-spin model that if a critical point for a nonzero $\epsilon$ persists, it is in the universality class of the RFIM even if the reference configurations are sampled at a fixed $T_0$. But this result goes beyond the $p$-spin model. More generally, the fate of the mean-field phase diagram of Fig. 12 in $2d$ and $3d$ is exactly the purpose of the present study and the outcome is shown in Fig. 10 (now Fig. 11).

In deriving the replicated effective action, the authors do not include mixed terms with both fields and field derivatives. Why?

In the spirit of the $\phi^4$-theory for studying the Ising critical universality class we have retained the minimum necessary terms in the effective Ginzburg-Landau action. The other terms are anyhow generated along the renormalization-group flow. The mixed terms involving fields and field derivatives then contribute to the nonzero anomalous dimension of the field and terms with replica fields and their derivatives describe the nonlocal (but still short-ranged) behavior of the disorder. We have added a comment saying that we neglect all these terms. A more detailed discussion is given in connection with the different types of disorder (see below).

Eq. (A33) is wrong on the second line. I believe it should be h.m

We thank the Referee for pointing out this mistake which has been corrected in the revised version of the manuscript.

After deriving the effective action the authors disregard all the sources of fluctuations but those in the random external field. Obviously in this way, one gets a random field Ising model, by definition. But instead of assuming the other fluctuating terms are irrelevant, the authors should show this irrelevance.

A short-range correlated random field $h(x)$ that breaks the $Z_2$ inversion symmetry in any given sample (the symmetry is only statistically recovered after disorder-averaging) and the 1-replica $\phi^4$-theory are the necessary ingredients for the RFIM universality class. The other disorder terms as well as additional gradient terms describing nonlocal but short-ranged behavior or terms associated with higher-order cumulants of the disorder are indeed generated along the renormalization-group flow, even in the standard RFIM (see, $\textit{e.g.}$, [Tarjus-Tissier, PRB (2008)]). This for the exact same reason that the whole Ising critical universality class can be described by starting from the Wilson-Ginzburg-Landau $\phi^4$-theory. We have added a paragraph of explanation in the Appendix A where this is discussed.

Minor points: - in Figure 10 some squares that should be empty look as filled because of the line passing through it - I would complement Figure 12 with a further figure showing the 4 colored curves together, so as to better show how the critical line gets reduced increasing $T_0$

We have done as suggested by the Referee.

In conclusion, we have clarified what is new in the present manuscript, added a subsection with results for $2d$ and $3d$ liquids for $T=T_0$, and improved the presentation of the analysis of the $p$-spin model. We have also presented additional data mostly for $2d$ systems.

Anonymous on 2022-01-09  [id 2082]

(in reply to Benjamin Guiselin on 2021-11-29 [id 1986])

I believe the authors have correctly improved the manuscript on the weakest points and have better rephrased some parts of the text.
In my opinion, this version is worth publishing.

---

## Editorial Decision

resubmitted